# Coupled Cluster con MōLe: Molecular Orbital Learning for Neural Wavefunctions

**Luca Thiede** [* 1 2]  **Abdulrahman Aldossary** [* 2 3]  **Andreas Burger** [1 2]  **Jorge Arturo Campos-Gonzalez-Angulo** [2 3]
**Alexander Zook** [4]  **Melisa Alkan** [4]  **Kouhei Nakaji** [4]  **Taylor Lee Patti** [4]  **Jérôme Florian Gonthier** [4]
**Mohammad Ghazi Vakili** [1 3]  **Alán Aspuru-Guzik** [1 2 3 4 5 6 7 8]

## Abstract

Density functional theory (DFT) is the most widely used method for calculating molecular properties; however, its accuracy is often insufficient for quantitative predictions. Coupled cluster (CC) theory is the most successful method for achieving accuracy beyond DFT and predicting properties that closely align with experiment. It is known as the "gold standard" of quantum chemistry. Unfortunately, the high computational cost of CC limits its widespread applicability. In this work, we present the Molecular Orbital Learning Model (MōLe), an equivariant machine learning model that directly predicts CC's core mathematical objects, the excitation amplitudes, from the mean-field Hartree-Fock molecular orbitals as inputs. We test various aspects of our model and demonstrate its very high data efficiency and remarkable out-of-distribution generalization to larger molecules and off-equilibrium geometries, despite being trained only on small equilibrium geometries. Finally, we also examine its ability to reduce the number of cycles required to converge CC calculations. MōLe can set the foundations for high-accuracy wavefunction-based ML architectures to accelerate molecular design and complement force-field approaches.

## 1 Introduction

Computational chemistry aims to predict the properties of molecules *in silico*, thereby facilitating insight into atomistic processes and advancing fields ranging from materials science to drug discovery. Density functional theory (DFT) has become the mainstream workhorse in computational chemistry, offering a good trade-off between speed and accuracy (Kohn & Sham, 1965; Szabo & Ostlund, 2012; Mardirossian & Head-Gordon, 2017). DFT solves a self-consistent field (SCF) problem to obtain the electron density, from which other properties can be derived. To further speed up DFT, recent work uses ML models to predict good initial guesses in the form of Hamiltonian matrices (Yu et al., 2023; Luo et al., 2025; Kim et al., 2025; Li et al., 2025), density matrices (Shao et al., 2023; Hazra et al., 2024; Febrer et al., 2025) or the electron density (Elsborg et al., 2025; Koker et al., 2024; Brockherde et al., 2017; Focassio et al., 2023; Jørgensen & Bhowmik, 2022; Rackers et al., 2023; Grisafi et al., 2018; Liu et al., 2025; Soares et al., 2025).

Aside from DFT acceleration, a rapidly growing class of Machine-Learned Interatomic Potentials (MLIPs) provides accurate (relative to DFT) energies and forces at low cost by learning to imitate DFT properties (Mehdizadeh & Schindler, 2025; Martyka et al., 2025; Chen & Dral, 2025; Banchode et al., 2025). However, even the best DFT functionals are limited in accuracy, in turn making MLIPs unreliable for applications that demand a high level of precision.

Thus, higher-accuracy ab initio methods are essential for reliably describing molecular properties. Among these methods, coupled cluster (CC) theory with single, double, and perturbative triples [CCSD(T)] is often considered the "gold standard" of quantum chemistry, achieving "chemical accuracy" (errors $\lesssim 1.6$ mHa) with respect to experimental data across various systems (Shavitt & Bartlett, 2009; Paldus et al., 1972; Eriksen et al., 2014; Helgaker et al., 2000). Unfortunately, the high computational cost of CCSD(T), which scales as $\mathcal{O}(N^7)$ with system size limits its routine application to small molecules. Non-perturbative versions of coupled cluster, either excluding triple excitations in CCSD or including triple excitations fully as in CCSDT, result in

*Equal contribution [1] Department of Computer Science, University of Toronto, Toronto, Canada [2] Vector Institute for Artificial Intelligence, Toronto, Canada [3] Department of Chemistry, University of Toronto, Toronto, Canada [4] NVIDIA, Toronto, Canada [5] Department of Materials Science & Engineering, University of Toronto, Toronto, Canada [6] Department of Chemical Engineering & Applied Chemistry, University of Toronto, Toronto, Canada [7] Acceleration Consortium, Toronto, Canada [8] Senior Fellow, Canadian Institute for Advanced Research (CIFAR), Toronto, Canada . Correspondence to: Alán Aspuru-Guzik <aspuru@nvidia.com>.

*Proceedings of the 43rd International Conference on Machine Learning*, Seoul, South Korea. PMLR 306, 2026. Copyright 2026 by the author(s).

scalings of $\mathcal{O}(N^6)$ and $\mathcal{O}(N^8)$, respectively.

To overcome the limitations of DFT and bring down the cost of correlated wavefunction methods, machine learning approaches explored learning from coupled cluster with property prediction with MLIPs (Smith et al., 2020; Ikeda et al., 2025; Messerly et al., 2025; O'Neill et al., 2025), effective hamiltonian matrix prediction (Tang et al., 2024), direct prediction with manually designed physics-inspired features with XGBoost and K-Nearest Neighbors (Townsend & Vogiatzis, 2020; 2019), neural DFT-functionals (Luise et al., 2025; Gao et al., 2024; Kanungo et al., 2025) as well as other beyond-DFT targets such as ML Green's-function representations (Venturella et al., 2024; 2025; Shang et al., 2025), active orbital prediction for selected CI (King et al., 2025) and neural networks as generalizing wavefunction representations (Gao & Günnemann, 2023; Scherbela et al., 2024) in variational quantum Monte Carlo.

The CC (de-)excitation amplitudes encode the complete correlated response of the electronic system. A model that learns these amplitudes can therefore recover the full CC property manifold. We would also expect amplitude prediction to be more data-efficient than direct structure-to-energy learning, as is done in MLIPs, which receive only a single scalar supervision signal per molecular geometry. By contrast, CC amplitudes provide a high-dimensional set of chemically structured targets that resolve how individual occupied and virtual orbitals correlate. Each molecule, therefore, supplies a fine-grained map of local correlation effects rather than only an aggregate energy label. For localized orbitals, this results in a spatially resolved training signal that gives a particularly intuitive explanation for the data efficiency. We discuss this in more detail in appendix C. In this work, we propose the Molecular Orbital Learning model (MōLe) that extends the idea of neural surrogates for densities to CC amplitudes. By lowering the complexity barrier of CC methods, we hope to expand their use and enable highly accurate property predictions across more applications than is currently possible. Our contributions are the following:

1. We design the first symmetry-aware neural architecture that takes molecular orbitals as input and outputs CC T-amplitudes.

2. We recalculate the QM7 (Rupp et al., 2012; Blum & Reymond, 2009) dataset at the CCSD/def2-SVP level of theory and several out-of-distribution datasets using out-of-equilibrium geometries and much larger systems compared to QM7.

3. We demonstrate that our model successfully predicts the CCSD amplitudes, resulting in energy errors of ~0.1 mHa, electron densities more accurate than MP2, and significantly fewer SCF iterations.

4. We show that, compared to MLIPs, even when trained with Δ-MP2-learning (Ramakrishnan et al., 2015), our model is more data efficient in- and out-of-distribution.

## 2 Theory

### 2.1 Hartree-Fock and molecular orbitals

In the Hartree-Fock (HF) approximation (Roothaan, 1951; Szabo & Ostlund, 2012), the many-electron wavefunction is represented by a single Slater determinant of orthonormal molecular orbitals (MOs) $\psi_p(\mathbf{r}|\{\mathbf{R}_A\})$, where $\{\mathbf{R}_A\}$ are the nuclear coordinates. Each MO $\psi_p(\mathbf{r}|\{\mathbf{R}_A\})$ is expanded in an atomic-orbital (AO) basis as

$$\psi_p(\mathbf{r}) = \sum_A \sum_{k \in \mathcal{K}_A} \sum_{\ell \in \mathcal{L}_{A,k}} \sum_{m=-\ell}^{\ell} C_{pA,k\ell m}\, \phi_{k\ell}^m(\underbrace{\mathbf{r} - \mathbf{R}_A}_{:=\mathbf{r}_A}).$$

Here, $A$ labels atoms with positions $\mathbf{R}_A$, while $k, \ell, m$ are the principal, azimuthal, and magnetic quantum numbers. The sets $\mathcal{K}_A$ and $\mathcal{L}_{A,k}$ refer to the element-dependent shells and sub-shells, respectively, included in the chosen basis on atom $A$, and $C_{pA,k\ell m}$ are the MO coefficients. There are usually as many molecular spatial orbitals as there are atomic orbitals $N_{\text{AO}}$. The first $N_{\text{electron}}$ molecular orbitals are called occupied, while the remaining $N_{\text{AO}} - N_{\text{electron}}$ are called virtual orbitals. Throughout the paper, we will use the letters $p$ and $q$ to index all orbitals; $i$ and $j$ refer to occupied orbitals, and $a$ and $b$ to virtual orbitals. $\phi_{k\ell}^m(\mathbf{r}_A) = R_k(|\mathbf{r}_A|)Y_\ell^m(\hat{\mathbf{r}}_A)$ are constructed from the pre-tabulated radial basis function $R_k(r)$ and the spherical harmonic $Y_\ell^m(\hat{\mathbf{r}})$.

To calculate the MO coefficients, we solve the Roothaan-Hall equations

$$\mathbf{F}(\mathbf{C})\mathbf{C} = \mathbf{C}\varepsilon, \tag{1}$$

with $\mathbf{F}(\mathbf{C})$ the Fock matrix, $\mathbf{C}$ the MO coefficient matrix, and $\varepsilon$ the diagonal matrix of orbital energies. Since $\mathbf{F}$ depends on $\mathbf{C}$, this equation is solved iteratively. The columns of $\mathbf{C}$ are eigenvectors, and therefore the MO coefficients are only determined up to an arbitrary sign. See Appendix B.1 for more details.

Intuitively, the Hartree-Fock wavefunction is the best approximate solution to the Schrödinger equation under the assumption that the electronic positions do not explicitly correlate with each other. To move past this limiting assumption, "post-Hartree-Fock" methods build on the Hartree-Fock wavefunction and introduce correlation in a systematically improvable manner. Møller-Plesset perturbation theory (MP$n$) and Coupled Cluster are among the most broadly used post-Hartree-Fock correlation methods.

## 2.2 Møller-Plesset-2 (MP2)

MP$n$ (Møller & Plesset, 1934; Szabo & Ostlund, 2012) is a perturbative expansion of the electronic correlation energy around the Hartree-Fock reference up to the $n$-th order. MP2 is the first nontrivial correction given by:

$$E_{\text{MP2}} = \frac{1}{4} \sum_{i,j,a,b} \underbrace{\frac{\langle ij||ab \rangle}{\varepsilon_i + \varepsilon_j - \varepsilon_a - \varepsilon_b}}_{t^{ab}_{ij,\text{MP2}}} \langle ij||ab \rangle, \quad (2)$$

with molecular orbital energies $\varepsilon_p$ and $\langle ij||ab \rangle$, the integral between orbitals $i, j, a, b$, see Appendix B.2 for background on integrals. MP2 formally scales as $\mathcal{O}(N^5)$ with a relatively small prefactor, since there is no need to solve an equation self-consistently.

## 2.3 Coupled cluster with single and double excitations

The coupled cluster method (Purvis III & Bartlett, 1982; Szabo & Ostlund, 2012) expresses the exact electronic wavefunction as an exponential excitation of a reference Hartree-Fock wavefunction:

$$|\Psi_{\text{CC}}\rangle = e^{\hat{T}} |\Phi_{\text{HF}}\rangle, \quad (3)$$

where the cluster operator $\hat{T} = \hat{T}_1 + \hat{T}_2 + \hat{T}_3 + \cdots$ generates single, double, triple, and higher excitations out of the reference state. Truncating $\hat{T}$ to a given excitation rank defines a hierarchy of systematically improvable methods: coupled cluster with singles and doubles (CCSD) includes single and double excitations, CCSDT adds triples, CCSDTQ adds quadruples, and so on. Therefore, in CCSD, the correlated wavefunction is

$$|\text{CCSD}\rangle = \exp\left(\hat{T}_1 + \hat{T}_2\right) |\Phi_{\text{HF}}\rangle, \quad (4)$$

with

$$\hat{T}_1 = \sum_{ia} t^a_i \, \hat{a}^\dagger_a \hat{a}_i, \quad \hat{T}_2 = \frac{1}{4} \sum_{ijab} t^{ab}_{ij} \, \hat{a}^\dagger_a \hat{a}^\dagger_b \hat{a}_j \hat{a}_i, \quad (5)$$

with $t^a_i$ and $t^{ab}_{ij}$ the singles and doubles amplitudes, and $\hat{a}$ and $\hat{a}^\dagger$ the Fermionic annihilation and creation operators. The CCSD amplitudes, $t^a_i$ and $t^{ab}_{ij}$, are the key objects which we usually obtain by solving a non-linear system of equations, see more in Appendix B.3. Solving the CCSD equations formally scales as $\mathcal{O}(N^6)$, with a large prefactor.

MP2 can be understood as the first-order approximation to coupled-cluster theory:

$$t^{ab}_{ij,\text{CCSD}} \approx t^{ab}_{ij,\text{MP2}}. \quad (6)$$

Since MP2 is computationally inexpensive compared to CCSD, it is typically used to initialize CCSD calculations.

An essential characteristic of the T-amplitudes, $t^a_i$ and $t^{ab}_{ij}$, is their asymptotic behavior as a function of the associated molecular orbitals separation: for two infinitely separated, closed shell molecular systems A and B, after localization (see Section 2.4), the molecular orbitals will be uniquely associated with either system A or B. Therefore, any excitation amplitude involving orbitals from both noninteracting subsystems A and B vanishes.

The CC amplitudes are all that is required to approximately calculate any electronic ground-state property of the system. Most notably, this includes the correlation energy

$$E_{\text{correlation}} = \sum_{ijab} \left( \frac{1}{4} t^{ab}_{ij} + \frac{1}{2} t^a_i t^b_j \right) \langle ij||ab \rangle. \quad (7)$$

This expression also holds for higher orders of non-perturbative CC methods, i.e., CCSDT, CCSDTQ, etc. Thus, we only ever need to predict the $T_1$ and $T_2$ tensors for energies, opening the door for our MōLe model to move beyond CCSD energies in the future.

## 2.4 Localized Molecular Orbitals

Canonical molecular orbitals (MOs), as defined in Equation (1), are generally delocalized over the entire molecule. We can transform MOs into a localized representation that leaves all observables invariant, see Appendix E. We hypothesize that locality represents an essential inductive bias (Chen et al., 2023) that facilitates easier training, which we experimentally confirm in Appendix E. Therefore, we use localized orbitals for all of our models.

## 3 Equivariance and equivariant models

Many mappings of physical properties are equivariant under coordinate-system transformations, meaning they transform predictably. Formally, a function $\boldsymbol{f} : \boldsymbol{X} \to \boldsymbol{Y}$ is equivariant with respect to the elements $g$ of a group $\mathcal{G}$, which acts on $\boldsymbol{X}$ and $\boldsymbol{Y}$ via $D_{\boldsymbol{X}}(g)$ and $D_{\boldsymbol{Y}}(g)$, if

$$\boldsymbol{f}(D_{\boldsymbol{X}}(g)\boldsymbol{x}) = D_{\boldsymbol{Y}}(g)\boldsymbol{f}(\boldsymbol{x}). \quad (8)$$

Oftentimes, the group $\mathcal{G}$ of interest is SO(3) the group of rotations in three-dimensional space, denoted by $\boldsymbol{Q} = (\alpha, \beta, \gamma)$, the Euler angles(Thomas et al., 2018; Geiger & Smidt, 2022). The irreducible representations (irreps) of this group are known as the Wigner-D matrices, with elements $D^{mm'}_\ell(\mathbf{Q})$, where the parameter $\ell$ is determined by the space they act upon.

For example, the Hartree-Fock algorithm is the map that assigns to each molecular geometry $\{\mathbf{R}\}$ a set of angular-momentum-labeled coefficient vectors that represent the molecular orbitals, $\mathbf{C}_{pA,k\ell}(\{\mathbf{R}\}) : \mathbb{R}^{3N_{\text{Atoms}}} \to \mathbb{R}^{2\ell+1}$. This

map is equivariant with respect to spatial rotations:

$$\mathbf{C}_{pA,k\ell}(D_1(\mathbf{Q})\{\mathbf{R}\}) = D_\ell(\mathbf{Q})\mathbf{C}_{pA,k\ell}(\{\mathbf{R}\}). \quad (9)$$

In contrast, the amplitudes $t_i^a$ and $t_{ij}^{ab}$ are invariant under rotations of the molecule:

$$t_{ij}^{ab}(D_1(\mathbf{Q})\{\mathbf{R}\}) = t_{ij}^{ab}(\{\mathbf{R}\}) \quad (10)$$

Another important symmetry is sign equivariance: If the signs of one of the molecular orbitals $i, j, a$ or $b$ associated with $t_{ij}^{ab}$ flips, the amplitude also flips its sign. Formally, for the amplitude $t_{ij}^{ab} = t(\mathbf{C}_i, \mathbf{C}_j, \mathbf{C}_a, \mathbf{C}_b)$ we have

$$t(-\mathbf{C}_i, \mathbf{C}_j, \mathbf{C}_a, \mathbf{C}_b) = t(\mathbf{C}_i, -\mathbf{C}_j, \mathbf{C}_a, \mathbf{C}_b) = ...$$
$$... = -t(\mathbf{C}_i, \mathbf{C}_j, \mathbf{C}_a, \mathbf{C}_b). \quad (11)$$

### 3.1 Equivariant Neural Networks

Equivariant neural networks are models that, by design, respect the equivariance property in Equation (8), often leading to greater data efficiency (Brehmer et al., 2025; Ngo & Ravanbakhsh, 2025; Suman et al., 2025). ML models for atomistic modeling are usually designed to respect the permutation of feature labels and O(3) symmetries by modeling the system as a graph neural network (GNN) with features $\mathbf{h}_{i,k,\ell,m}$ that carry the O(3) irreps, and therefore transform under rotation with the Wigner D-matrices. Two symmetry-adapted features, $\mathbf{p}_{\ell_1}$ and $\mathbf{g}_{\ell_2}$, can interact with each other while preserving equivariance using the bilinear Clebsch-Gordan tensor product (Thomas et al., 2018)

$$\mathbf{h}_{\ell_3 m_3} = \sum_{m_1=-\ell_1}^{\ell_1} \sum_{m_2=-\ell_2}^{\ell_2} \mathcal{C}_{\ell_1 m_1, \ell_2 m_2}^{\ell_3 m_3} \mathbf{p}_{\ell_1 m_1} \mathbf{g}_{\ell_2, m_2}, \quad (12)$$

where $\mathcal{C}_{(\ell_1, m_1),(\ell_2, m_2)}^{(\ell_3, m_3)}$ is the Clebsch-Gordan coefficient that implements the O(3)-equivariant coupling of the irreps $\ell_1$ and $\ell_2$ into the output irrep $\ell_3$, mapping basis elements $(\ell_1, m_1)$ and $(\ell_2, m_2)$ to $(\ell_3, m_3)$. Equation (12) is used to build equivariant message passing from irrep-carrying node and edge features. See Appendix A for more details.

The MACE architecture (Batatia et al., 2022) is one of the most prominent equivariant message-passing architectures and will be an important component of our model design; see Section 4.2.2.

## 4 MōLe architecture

Our MōLe architecture design is guided by satisfying MO and CC wavefunction symmetries and asymptotics:

1. MO coefficients are rotation equivariant, Equation (9)

2. T-amplitudes are rotation invariant, Equation (10)

3. T-amplitudes are sign equivariant, Equation (11)

4. The T-amplitudes coupling the localized MOs of two infinitely-separated closed-shell molecules are 0.

An overview of the model architecture is shown in Figure 1, and details are provided in Figure 6. The model proceeds in four main steps: First, we run a Hartree-Fock calculation and localize the MOs. We then "embed" the coefficients onto "graph-states", one graph per MO, by initializing equivariant GNN features with the MO coefficients. The features are then processed by $(T)$ transformer blocks, each combining a MACE-like layer, a layer norm, and an attention mechanism to capture local atomic and long-range orbital correlations. Finally, a readout layer converts the latent MO features into $T_1$ and $T_2$ amplitudes.

### 4.1 MO embedding

Since different atom types have different numbers of basis functions, but GNNs expect all atoms to have the same number of features, we need to pad the MO coefficients before we can use them as initial features in our equivariant GNN, see Figure 1 for an illustration and Appendix F.1 for details.

Using an equivariant linear layer $\mathbf{W}_{\ell,k\bar{k}}$, the initial MO coefficients are then embedded into a hidden dimension $K, k \in \{1, ..., K\}$ that will be kept throughout the transformer blocks:

$$\mathbf{x}_{pA,k\ell m}^{(0)} = \sum_{\bar{k}} W_{\ell,k\bar{k}}^{(0)} C_{pA,\bar{k}\ell m}^{\text{padded}}, \quad (13)$$

where $\mathbf{W}_{\ell,k\bar{k}}$ are learnable equivariance preserving weight matrices. Note that, in contrast to machine learning force fields where a single graph neural network maps to a target property, here, there are $N_{MO}$ graph neural networks in parallel, one for each MO, with the same weights shared across them.

### 4.2 Equivariant Transformer block

Next, an equivariant transformer block processes the MO embeddings across multiple transformer layers, where each layer transforms the input features while preserving their shape, rotation and sign equivariance, and only makes MOs interact if they belong to the same fragment. The $t$-th layer transforms the input features $\mathbf{x}_{pA}^{(t)} = \{x_{pA,k\ell m}^{(t)}\}$ to produce output features $\mathbf{x}_{pA}^{(t+1)}$ of the same shape. Throughout the paper, we use bold letters to denote arrays without specifying their indices.

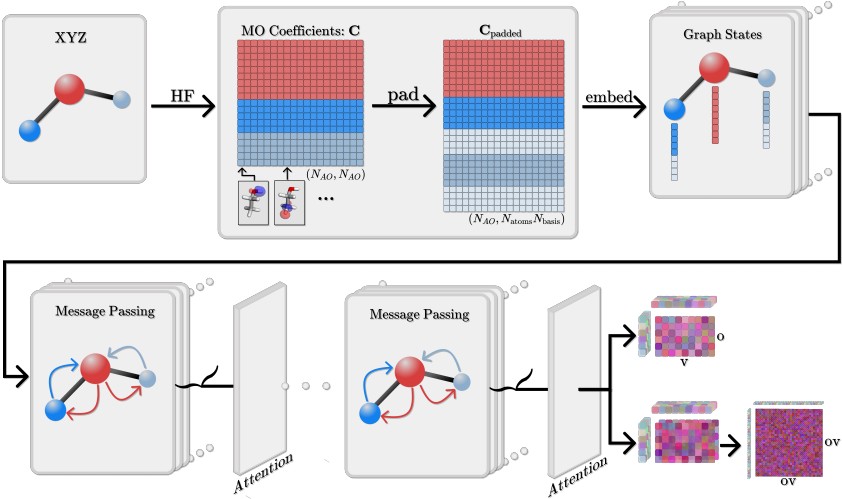

*Figure 1.* Given a molecule, a Hartree-Fock calculation provides the molecular orbitals represented by their coefficient matrix $\mathbf{C}$. The coefficient vector is padded for each atom to ensure that they all have the same number of basis coefficients, enabling their embedding in an equivariant neural network. The model then alternates message passing to mix information within the MOs and attention layers to mix information between MOs. Finally, the embeddings are read out by "outer product-like" operations, outputting the $T_1$ and $T_2$ amplitudes.

The transformer consists of three main components: A MO-Attention block to capture correlation between MOs, a normalization layer, and "Odd-MACE" to mix features within an MO, interleaved with skip connections as shown in Figure 6.

### 4.2.1 MO-ATTENTION

The MO-Attention mechanism processes the MO embeddings through multi-head attention, where each head applies learnable weight matrices to construct the query (Q), key (K), and value (V) representations. For each attention head $h$, the weight matrices $\mathbf{W}^{(Q,h,t)}$, $\mathbf{W}^{(K,h,t)}$, and $\mathbf{W}^{(V,h,t)}$ project $x_{pA,k\ell m}^{(t)}$ to the key, query and values features:

$$Q_{pA,k\ell m}^{(h,t)} = \sum_{\bar{k}} W_{\ell,k\bar{k}}^{(Q,h,t)} x_{pA,\bar{k}\ell m}^{(t)}, \tag{14}$$

and similarly for K and V. Next, we normalize the $L_2$ norm of the features:

$$\tilde{Q}_{pA,k\ell m}^{(h,t)} = \mathrm{Norm}_\epsilon(\mathbf{Q}_p^{(h,t)}), \quad \tilde{K}_{pA,k\ell m}^{(h,t)} = \mathrm{Norm}_\epsilon(\mathbf{K}_p^{(h,t)})$$

with

$$\mathrm{Norm}_\epsilon(\mathbf{X}_p^{(h,t)}) = \frac{X_{pA,k\ell m}^{(h,t)}}{\sqrt{\sum_{Ak\ell m} |X_{pA,k\ell m}^{(h,t)}|^2 + |\epsilon|}}, \tag{15}$$

where $\epsilon$ is learnable. The attention score $S_{p,q}^{(h,t)}$ between MO $p$ and MO $q$ is computed as:

$$S_{pq}^{(h,t)} = \sum_{Ak\ell m} \tilde{Q}_{pA,k\ell m}^{(h,t)} \cdot \tilde{K}_{qA,k\ell m}^{(h,t)}. \tag{16}$$

To preserve sign equivariance, we skip the usual softmax normalization and directly sum and renormalize the attention-weighted contributions, which is structurally similar to previously proposed transformer variants like the TransNormer(Qin et al., 2022):

$$\tilde{o}_{pA,k\ell m}^{(h,t)} = \sum_q \tilde{S}_{pq}^{(h,t)} V_{qA,k\ell m}^{(h,t)}, \quad o_{pA,k\ell m}^{(h,t)} = \mathrm{Norm}_\epsilon(\tilde{\mathbf{o}}_p^{(h,t)})$$

Finally, we sum over the different heads:

$$x_{pA,k\ell m}^{(t+1)} = \sum_h w_h^{(t)} o_{pA,k\ell m}^{(h,t)},$$

where $w_h$ are learnable parameters. The updated node features $\mathbf{x}_{pA}^{(t+1)}$ are then passed to the "Odd-MACE" block.

Since the attention only uses inner products between features with the same $\ell$, which is rotation invariant, and a $L_2$ norm, which is rotation invariant, the attention weights are also rotation invariant. Summing the rotation-equivariant features with invariant weights therefore preserves rotation equivariance of the attention mechanism. Sign equivariance is preserved because the inner product is sign equivariant with respect to both MO features. Finally, if two MOs are localized on two non-interacting fragments, their inner product is 0, since there are no overlapping non-zero coefficients. Therefore, the attention mechanism preserves size-extensivity.

### 4.2.2 ODD-MACE

To allow mixing within each MO feature, one MACE layer is applied after each attention block. MACE is an equivariant neural network, where the key equivariant non-linear operation can be viewed as a tensor polynomial in the irreps-carrying features,

$$\mathbf{x}_{p,A,k}^{(\nu)} = \sum_{\xi=0}^{\nu} \sum_{k'} \mathbf{W}_{k,k'}^{(\xi)} \left(\mathbf{x}_{p,A,k'}\right)^{\otimes \xi}. \qquad (17)$$

The tensor power $(\mathbf{h}_{p,k'})^{\otimes \xi}$ denotes the stack of all order-$\xi$ equivariant tensor products constructed by repeated application of Equation (12) to the components of $\mathbf{x}_{p,k'}$, and $\mathbf{W}_{k,k'}^{(\xi)}$ is a learned linear map. The tensor polynomial is applied to the node features after each message-passing layer. In practice, Equation (17) is implemented efficiently using Horner's method. It is important to note that the even tensor monomials are sign invariant, while the odd tensor monomials are sign equivariant. Thus, to enforce sign equivariance in our model, we restrict the monomial order $\xi$ in Equation (17) to the odd ones, which we call "Odd-MACE".

Since MACE is an equivariant neural network, it preserves the rotational equivariance of the MO features. Message passing itself is sign equivariant with respect to its input features, such that together with the restriction to odd monomials, Odd-Mace ensures sign equivariance. Finally, the cutoff radius enforced in message passing prevents interactions between far-separated fragments, thereby ensuring size extensivity.

### 4.3 Layer Normalization

Our architecture interleaves equivariant layer normalization blocks between each attention and an Odd-MACE block, as shown in Figure 6. The layer normalization is inspired by (Liao et al., 2023), however, we found it beneficial to make $\epsilon$ learnable. Please see Appendix F.2 for details.

### 4.4 Readout

The readout layers transform the latent MO features into the final $T_1$ and $T_2$ amplitudes.

### 4.4.1 $T_1$ READOUT

The $T_1$ readout combines features from two molecular orbitals (one occupied orbital $i$ and one virtual orbital $a$) to produce single excitation amplitudes. We start by normalizing each feature using:

$$\tilde{x}_{iA,k\ell m}^{(T)} = \text{Norm}_\epsilon(\mathbf{x}_i^{(T)}). \qquad (18)$$

Using an element-wise tensor product, we calculate invariant pairwise features for every $\ell$:

$$\tilde{y}_{iA,\bar{k}\ell m}^{(T)} = W_{\ell,,k\bar{k}}^{\text{single}} \tilde{x}_{iA,\bar{k}\ell m}^{(T)}, \quad \tilde{y}_{aA,\bar{k}\ell m}^{(T)} = W_{\ell,k\bar{k}}^{\text{single}} \tilde{x}_{aA,\bar{k}\ell m}^{(T)}$$

$$y_{ia,k,\ell} = \sum_{A,m_1,m_2} \tilde{y}_{iA,\bar{k}\ell m_1}^{(T)} \tilde{y}_{aA,\bar{k}\ell m_2}^{(T)} \mathcal{C}_{\ell,m_1,\ell,m_2}^{00}, \qquad (19)$$

where $W_{\ell,k\bar{k}}^{\text{single}}$ are learnable weights. Finally, a sign equivariant "Odd-MLP" (no bias and tanh nonlinearity) processes the features to predict the final amplitudes:

$$t_i^a = \text{Odd-MLP}(\mathbf{y}_{ia}). \qquad (20)$$

### 4.4.2 $T_2$ READOUT

The $T_2$ readout combines features from two occupied orbitals $i$ and $j$, and two virtual orbitals $a$ and $b$, to produce double excitation amplitudes. First, we calculate intermediate pair-features:

$$\tilde{f}_{iaA,k\ell m} = \sum_{\ell_1 m_1 \ell_2 m_2} w_{\bar{k}\ell\ell_1\ell_2}^{(1)} x_{iA,k\ell_1 m_1}^{(T)} x_{aA,\bar{k}\ell_2 m_2}^{(T)} \mathcal{C}_{\ell_1 m_1 \ell_2 m_2}^{\ell m},$$

$$f_{iaA,k\ell m} = \text{Norm}_\epsilon(\tilde{\mathbf{f}}_{ia}),$$

$$f_{iaA,k\ell m}' = \sum_{\bar{k}} W_{\ell,k\bar{k}} f_{iaA,\bar{k}\ell m}$$

and similar for the $j, b$ orbitals, resulting in $f_{jbA,k\ell m}$ These intermediate representations are then further combined to create the final four-orbital features:

$$y_{ijab,k\ell} = \sum_A \sum_m f_{iaA,k\ell m}' f_{jbA,k\ell m}' C_{\ell,m,\ell,m}^{00}, \qquad (21)$$

used to predict the amplitudes with an Odd-MLP:

$$t_{ij}^{ab} = \text{Odd-MLP}(\mathbf{y}_{ijab}) + t_{ij,\text{MP2}}^{ab}, \qquad (22)$$

Since we need the two-particle integrals for evaluating the energies anyway (see Equation (7)), we get the MP2 amplitudes for free. Therefore, we train our model to predict only the difference between MP2 and CCSD amplitudes, a technique denoted by $\Delta$-MP2-learning (Ramakrishnan et al., 2015).

## 5 Experiments

Our experimental goal is to assess the efficacy of our neural network surrogate for coupled-cluster (CC) amplitudes in predicting corrections from MP2 to CCSD amplitudes and whether these predictions yield strong performance on downstream tasks. Fundamentally, our thesis is that CC methods are too expensive to afford large-scale datasets on large molecules. This is already true for CCSD, but it would

be even more severe if we moved to higher-order CC methods, such as CCSDT. Therefore, we will restrict ourselves to training on the relatively small QM7 dataset. We are then interested in six aspects of our model:

1. In-distribution accuracy: How well does MōLe reproduce CCSD energies on the QM7 test split?

2. Size extrapolation: How well does the model generalize to molecules significantly larger than those in the training distribution?

3. Off-equilibrium extrapolation: Can the model remain accurate for off-equilibrium geometries, even though it is trained only on equilibrium structures?

4. Ultra low data regime: How well does MōLe perform when trained on 100 samples only?

5. Other properties: To what extent do amplitude predictions support accurate prediction of other observables, such as the electron density?

6. SCF convergence: Can the predicted amplitudes serve as a good initial guess, reducing CCSD solver iterations?

Given the restriction to small datasets, we are particularly interested in the data efficiency of our model. To contextualize the data efficiency, we compare it against the MACE, eSEN (Fu et al., 2025), TensorNet (Simeon & De Fabritiis, 2023), and GemNet (Gasteiger et al., 2021) MLIP architectures. For fairness, we train the MLIPs in a $\Delta$-learning setting, where we predict only the correction from MP2 to CCSD energies. We also train a model without delta learning to add context for the results one would obtain with standard MACE training. We emphasize that MLIPs and amplitude-based models target different objectives: MLIPs prioritize speed and scalability, whereas MōLe is designed for data efficiency, completeness of properties, and CC-compatibility. Therefore, the comparison should be interpreted as a reference point rather than a claim that MōLe is a universal replacement for MLIPs.

## 5.1 QM7 experiments

QM7 consists of 7165 small organic molecules composed of C, N, O, S, and H. We use geometries provided in the original paper(Rupp et al., 2012; Blum & Reymond, 2009) and recompute all labels at the CCSD/def2-SVP(Weigend & Ahlrichs, 2005) level of theory. For each molecule, we store the full set of coupled-cluster amplitudes. We train a model on an 80/20 split, and additionally, three models on 100-molecule subsets using three random seeds. Hyperparameters and training details for MōLe and the MLIPs are provided in Appendix I. We are using the checkpoint

obtained from training on QM7 for all the following evaluations.

The mean absolute error (MAE) on the QM7 test set for the $T_1$ amplitudes is $6.5 \times 10^{-5}$ and $9.84 \times 10^{-7}$ for the $T_2$ amplitudes. This translates to an excellent energy error of 0.12 mHa, slightly better than the $\Delta$-MP2 MLIPs (see table 1) and significantly better than the non $\Delta$-learning MACE model. Surprisingly, even on the much smaller 100 molecule subset, MōLe achieves strong energy predictions of 0.66 mHa. In this low-data regime, the data efficiency gap to the MLIPs becomes much clearer. As an example, we plot the prediction of MōLe and the ground truth amplitudes in Figure 7, illustrating that MōLe successfully predicts the highly non-trivial amplitude pattern. Next, we evaluate our model on several out-of-distribution datasets, each of which we generate at CCSD/def2-svp level of theory.

## 5.2 Size extrapolation experiments

To study the size generalization, we evaluate MōLe on two out-of-distribution datasets containing molecules twice the size of those in QM7. This is particularly important given the steep scaling cost of CC methods, rendering the generation of datasets with large molecules prohibitive.

**Amino Acids** The amino acids dataset contains 18 structurally diverse amino acids with up to 15 heavy atoms.

**PubChem** We further construct a diverse benchmark by randomly sampling 100 molecules from the PubChem database, containing 14 heavy atoms made from C, H, N, and O. The PubChem test set represents the most diverse and chemically challenging benchmark in our evaluation.

Table 1 shows that MōLe generalizes better to larger molecules, in particular in the ultra-low data regime.

## 5.3 Off equilibrium experiments

We also test the model's ability to generalize to off-equilibrium geometries, despite being trained only on equilibrium structures, using three distinct chemical systems:

**Diels-Alder reaction** The Diels-Alder reaction is one of the most commonly studied reactions in chemistry. As an example, we use the Diels-Alder reaction path, turning ethylene and 1,3-butadiene into cyclohexene. The energy error along the intrinsic reaction coordinate is shown in Figure 2a.

**Dihedral Scan** Next, we perform a dihedral scan of the butane molecule, as it forces the geometry through a transition point. The energy error and zero-shifted relative energy scan is plotted in Figure 2b.

**Chair-To-Boat** The chair-to-boat transition of cyclohexane is a famous example of conformational isomerism, where

*Table 1.* Mean absolute error of energy predictions in mHa. "MACE ($\Delta$-MP2)", "eSEN ($\Delta$-MP2)", "TensorNet ($\Delta$-MP2)" and "GemNet ($\Delta$-MP2)" models are trained to predict $\Delta E = E_{\text{CCSD}} - E_{\text{MP2}}$ while "MACE" is trained to predict $E_{\text{CCSD}}$ directly. MōLe was trained to predict $\Delta t_{ij}^{ab} = t_{ij,\text{CCSD}}^{ab} - t_{ij,\text{MP2}}^{ab}$. We train each model on a 100 molecule subset (denoted "Model"-100), and on the full training split (5732 molecules) of QM7 (denoted "Model"). The 100 molecule models are averaged over three seeds. Pure MACE without MP2 scales square up to a certain system size, and then linear due to cutoff radii. Similar cutoffs can be made for wavefunction methods as well but are harder to implement, which is why we list them as the fifth power here.

| Model | Complexity | Train set | Size extrapolation | | Out-of-equilibrium | | |
| --- | --- | --- | --- | --- | --- | --- | --- |
| | | QM7 | Amino acids | PubChem | Diels-Alder | Dihedral sc. | Chair-to-boat |
| MACE-100 | $\mathcal{O}(N)$–$\mathcal{O}(N^2)$ | 19.15 | 26.41 | 64.89 | 15.01 | 15.37 | 26.56 |
| MACE-100 ($\Delta$-MP2) | $\mathcal{O}(N^5)$ | 1.64 | 5.53 | 5.29 | 3.17 | 0.79 | 1.16 |
| eSEN-100 ($\Delta$-MP2) | $\mathcal{O}(N^5)$ | 1.48 | 7.43 | 17.41 | 5.55 | 2.74 | 2.53 |
| TensorNet-100 ($\Delta$-MP2) | $\mathcal{O}(N^5)$ | 6.60 | 12.16 | 16.81 | 11.33 | 9.12 | 5.66 |
| GemNet-100 ($\Delta$-MP2) | $\mathcal{O}(N^5)$ | 2.52 | 5.33 | 9.19 | 3.74 | 0.76 | 6.07 |
| MōLe-100 | $\mathcal{O}(N^5)$ | **0.66** | **0.67** | **2.80** | **1.50** | **0.33** | **0.24** |
| MACE | $\mathcal{O}(N)$–$\mathcal{O}(N^2)$ | 1.83 | 10.60 | 17.74 | 7.05 | 4.61 | 4.50 |
| MACE ($\Delta$-MP2) | $\mathcal{O}(N^5)$ | 0.16 | 0.49 | 2.24 | 1.57 | 0.35 | 0.39 |
| eSEN ($\Delta$-MP2) | $\mathcal{O}(N^5)$ | 0.13 | 1.56 | 4.66 | 1.15 | 0.59 | 0.66 |
| TensorNet ($\Delta$-MP2) | $\mathcal{O}(N^5)$ | 0.21 | 1.42 | 3.00 | 1.18 | 0.44 | 0.35 |
| GemNet ($\Delta$-MP2) | $\mathcal{O}(N^5)$ | 0.44 | 2.07 | 7.95 | 1.78 | **0.16** | 0.19 |
| MōLe | $\mathcal{O}(N^5)$ | 0.12 | 0.78 | 1.63 | 1.16 | 0.22 | **0.08** |
| MōLe (256) | $\mathcal{O}(N^5)$ | **0.06** | **0.43** | **1.10** | **0.96** | 0.41 | 0.43 |

cyclohexane undergoes a "ring flip" by rotating its carbon-carbon single bonds. This process forces the molecule through a high-energy half-chair transition state and a local minimum twist-boat before reaching the boat conformation.

Table 1 shows that MōLe has, except for the Diels-Alder reaction, where it is tied with eSEN, significantly lower error for each system, again particularly in the ultra low data regime.

### 5.4 One-particle properties

From the amplitudes, we can derive the one-particle reduced density matrix (1-RDM) $\gamma(\mathbf{r}, \mathbf{r'}) = \gamma_{pq}\psi_p(\mathbf{r})\psi_q(\mathbf{r'})$ with XCCSD (Korona & Jeziorski, 2006), see Appendix D. From the 1-RDM, we can derive any one-particle property, including spatially resolved ones. The most fundamental of these is the electron density $\rho(\mathbf{r}) = \gamma(\mathbf{r}, \mathbf{r})$, which in turn lets us calculate other spatially resolved quantities like the electron localization function (ELF) (Becke & Edgecombe, 1990), Fukui function (Faver & Merz, 2010; Razali et al., 2024), Bader's QTAIM (Bader, 1990) and X-ray diffraction patterns. We report the Frobenius norm of the density matrix error in Table 2, which shows that our density matrices are much better than MP2. As an example, we plot the error of the electron density $|\rho_{\text{Model}}(\mathbf{r}) - \rho_{\text{CCSD}}(\mathbf{r})|$ in Figure 3. We see that MōLe performs significantly better than MP2.

### 5.5 CCSD cycle reduction

In cases where the MōLe model is not fully trusted, for example, for unusual molecular structures, the guarantees provided by rigorous theory can be desirable. In that case, the

*Table 2.* The Frobenius norm of the error between the density matrix of MP2 and MōLe.

| Model | Amino Acids | PubChem |
| --- | --- | --- |
| MP2 | 0.59 | 0.90 |
| MōLe | **0.13** | **0.36** |

predicted MōLe amplitudes can serve as an initial guess for a CCSD solver to reduce the number of cycles needed for convergence. We test this on the out-of-distribution molecules from the Amino Acids and PubChem datasets. We turn the Direct Inversion of Iterative Subspace (Scuseria et al., 1986) off to save memory and set our convergence threshold to $10^{-3}$ for the $L_2$ norm of the T-amplitude changes. This threshold corresponds to energy errors $|E - E_{\text{tight}}| \lesssim 0.1$ mHa compared to a very tightly converged calculation. The amplitudes predicted by MōLe result in 40-50% reduction of CCSD solver iterations, the exact numbers are in Table 5 in the appendix. Notably, for the PubChem molecules, three out of one hundred systems failed to converge with the default MP2 guess, two of which did converge with our predicted amplitudes, demonstrating the high quality and practical utility of MōLe's predictions.

### 5.6 Impact of model scaling and ablation study

We investigate the performance of our model as a function of transformer depth and width. In particular, we scale the transformer depth from 1 to 4 layers and its width from 32 to 128 irreps. We train the models for one week on the QM7 training set. The model's $T_2$ error on the QM7 test set is plotted in Figure 4. Our model's performance improves

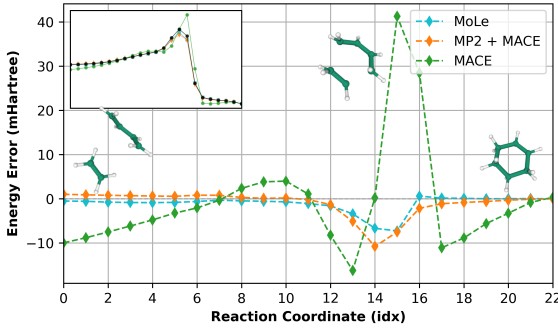

*(a)* Diels-Alder reaction of ethene and 1,3-butadiene turning into cyclohexene.

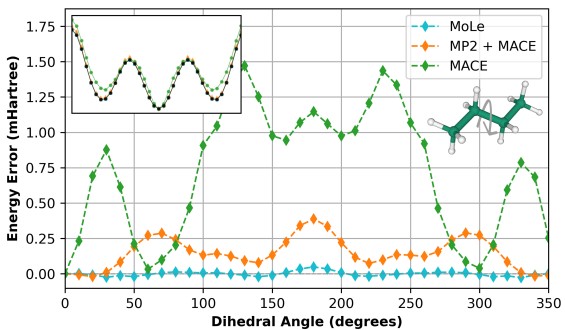

*(b)* Dihedral scan of the butane molecule.

*Figure 2.* The energy error of MōLe, MP2+MACE (i.e., Δ-MP2), and MACE along two scans. In the inset, the potential energy surface is shown, with the black line indicating the ground truth energies. MōLe achieves lower error particularly for the transition state region, while MACE would overestimate the activation energies.

monotonically with increasing transformer size, suggesting that the bottleneck is mainly capacity rather than data efficiency, and that scaling could therefore improve results further. We further perform an ablation study where we remove the attention and Odd-MACE from the model, see Table 4. We see that the validation loss suffers dramatically from removing both the gnn or attention part.

### 5.7 Time complexity

CCSD has a formal time complexity of $\mathcal{O}(N^6)$, while MōLe should have a theoretical time complexity of $\mathcal{O}(N^5)$, as there are $\mathcal{O}(N^4)$ many amplitudes, each costing $\mathcal{O}(N)$ to compute. To validate these expectations empirically, we timed increasingly larger alkane systems (C13-C21) and fit a power law for HF, MP2, CCSD, localization, and MōLe. See Figure 8 in the appendix for the scaling plot. Empirical scaling for MōLe is $\mathcal{O}(N^{3.3})$, indicating a large prefactor in the attention layer before the readout $\mathcal{O}(N^5)$ scaling, while CCSD gives $\mathcal{O}(N^{5.9})$ in line with our expectations. For

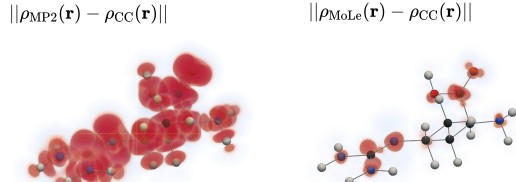

*Figure 3.* The electron density error of MP2 and MōLe on L-Arginine amino acid. The error is plotted at the 95% percentile of the MP2 error.

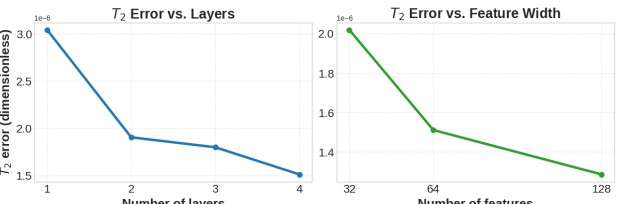

*Figure 4.* Left: Scaling the transformer's depth monotonically decreases the prediction error up to four layers. Right: Scaling the transformer width also decreases the prediction error monotonically.

these systems, MōLe is $\sim 1000\text{x}$ faster than CCSD when compared on a single GPU. These results demonstrate that MōLe adds negligible compute overhead on top of MP2 while recovering CCSD-level energies, with a much more favorable scaling than CCSD.

## 6 Conclusions and Outlook

In this work, we presented the first equivariant neural architecture to predict CC amplitudes from molecular orbital inputs. We trained our model on QM7 at the CCSD/def2-SVP level of theory and evaluated it on a diverse set of datasets and tasks, ranging from energies to electron densities and cycle reductions. The model performs well on both the QM7 test set and several out-of-distribution splits, achieving significantly higher data efficiency than MLIPs. Since higher levels of CC theory also need only the $T_1$ and $T_2$ amplitudes to compute energies, our work lays the groundwork to learn from very high levels of theory (e.g. CCSDT) with very high data efficiency. Future work will include sparse amplitude prediction to escape the cubic scaling, larger basis sets, and prediction of the lambda tensor for response properties.

## Impact Statement

This work aims to advance machine learning methods for high-accuracy molecular simulation by predicting coupled-cluster amplitudes more efficiently. More accurate and data-efficient electronic-structure surrogates could benefit applications such as catalyst discovery, materials design, drug discovery, and the study of molecular properties that are otherwise computationally expensive to obtain.

At the same time, methods that accelerate molecular simulation can have dual-use implications. In particular, improved access to high-accuracy predictions could lower the computational barrier for screening molecules with hazardous properties, including energetic materials or toxic compounds. Our contribution is methodological and does not target such applications, does not provide a generative design pipeline, and is evaluated on small organic molecules and benchmark molecular geometries rather than on hazardous chemical spaces. Nevertheless, we recognize that future extensions of this line of work could be incorporated into broader molecular design workflows with non-trivial safety considerations.

Responsible deployment of such models should therefore include application-specific risk assessment, restrictions on use in hazardous chemical design contexts, careful dataset curation, and evaluation of downstream workflows before release or operational use. We view these considerations as important for future work on scalable wavefunction-based machine learning models.

## Acknowledgment

L.T. and A.B. acknowledge the AIST support to the Matter lab for the project titled "SIP project - Quantum Computing". A.A. gratefully acknowledges King Abdullah University of Science and Technology (KAUST) for the KAUST Ibn Rushd Postdoctoral Fellowship. J.A.C.-G.-A. acknowledges funding of this project by the National Sciences and Engineering Research Council of Canada (NSERC) Alliance Grant #ALLRP587593-23 (Quantamole) and also acknowledges support from the Council for Science, Technology and Innovation (CSTI), Cross-ministerial Strategic Innovation Promotion Program (SIP), "Promoting the application of advanced quantum technology platforms to social issues" (Funding agency: QST). A.A.-G. thanks Anders G. Frøseth for his generous support. A.A.-G. also acknowledges the generous support of Natural Resources Canada and the Canada 150 Research Chairs program. Resources used in preparing this research were provided, in part, by the Province of Ontario, the Government of Canada through CIFAR, and companies sponsoring the Vector Institute. This research is part of the University of Toronto's Acceleration Consortium, which receives funding from the CFREF-2022-00042 Canada First Research Excellence Fund. This research was enabled in part by support provided by SciNet and the Digital Research Alliance of Canada (`alliancecan.ca`).

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

# A  Equivariant graph neural networks

SO(3)-Equivariant graph neural networks (GNNs) currently represent the most successful class of machine-learning models for interatomic potentials. In these approaches, a molecule is represented as a graph $\mathcal{G} = (\mathcal{V}, \mathcal{E})$ embedded in three-dimensional space $\mathbb{R}^3$, where nodes $I \in \mathcal{V}$ correspond to atoms located at positions $\mathbf{r}_I \in \mathbb{R}^3$, and edges $(A, B) \in \mathcal{E}$ are defined by a distance cutoff. At layer $t$, each node carries a feature vector $\mathbf{h}_A^{(t)}$ made up of a direct sum of irreducible representations (irreps) of SO(3),

$$\mathbf{x}_A^{(t)} = \bigoplus_{\ell=0}^{\ell_{\max}} \mathbf{x}_{A,\ell}^{(t)}, \qquad \mathbf{x}_{A,\ell}^{(t)} = \{\mathbf{x}_{A,\ell,m}^{(t)}\}_{m=-\ell}^{\ell}, \tag{23}$$

where each order-$\ell$ array has $2\ell + 1$ components. Feature updates are performed by exchanging messages along edges, aggregating them in a permutation-invariant manner (typically by summation or attention-weighted summation), and combining the result with the target node's current features.

In SO(3)-equivariant GNNs, node features transform covariantly under a global rotation $\mathbf{Q}$ according to the Wigner $\mathbf{D}_\ell(\mathbf{Q})$-matrices,

$$\mathbf{x}_{A,\ell,m}^{(t)}\big(\mathbf{D}_1(\mathbf{Q})\{\mathbf{r}_k\}_{k=1}^N\big) = \sum_{m'=-\ell}^{\ell} \mathbf{D}_{\ell,mm'}(\mathbf{Q})\, \mathbf{h}_{A,\ell,m'}^{(t)}\big(\{\mathbf{r}_k\}_{k=1}^N\big). \tag{24}$$

Translation equivariance is ensured by constructing messages only from relative displacements $\mathbf{r}_{A,B} = \mathbf{r}_B - \mathbf{r}_A$. If full O(3) equivariance is desired, each order-$\ell$ feature additionally carries a parity label specifying its behavior under spatial inversion.

Couplings between irreps $\mathbf{p}_{\ell_1}$ and $\mathbf{g}_{\ell_2}$ are formed via the Clebsch-Gordan tensor product (Thomas et al., 2018),

$$\mathbf{x}_{\ell_3,m_3} = \Big(\mathbf{p}_{\ell_1,m_1} \otimes_{\ell_1,\ell_2}^{\ell_3} \mathbf{g}_{\ell_2,m_2}\Big)_{m_3} = \sum_{m_1=-\ell_1}^{\ell_1} \sum_{m_2=-\ell_2}^{\ell_2} \mathbf{C}_{(\ell_1,m_1),(\ell_2,m_2)}^{(\ell_3,m_3)} \mathbf{p}_{\ell_1,m_1} \mathbf{g}_{\ell_2,m_2}. \tag{25}$$

We use this to construct messages sent from a source atom $B$ to a target atom $A$ as

$$\mathbf{v}_{A,B,\ell_3} = \mathbf{v}_{\ell_3}(\mathbf{x}_A, \mathbf{x}_B, \mathbf{r}_{A,B}) = \sum_{\ell_1,\ell_2} w_{\ell_1,\ell_2,\ell_3}(||\mathbf{r}_{A,B}||) \left( \mathbf{f}_{\ell_1}(\mathbf{x}_A, \mathbf{x}_B) \otimes_{\ell_1,\ell_2}^{\ell_3} \mathbf{Y}_{\ell_2}\left( \frac{\mathbf{r}_{A,B}}{||\mathbf{r}_{A,B}||} \right) \right), \tag{26}$$

where $\mathbf{Y}_{\ell_2}$ denotes the $(2\ell_2 + 1)$-dimensional vector of spherical harmonics of degree $\ell_2$ and $w_{\ell_1,\ell_2,\ell_3} : \mathbb{R} \to \mathbb{R}$ is a learned radial weighting function, usually implemented as a neural network. The function $\mathbf{f}_{\ell_1}$ is a simple map combining node features (which reduces to $\mathbf{f}_{\ell_1}(\mathbf{h}_I, \mathbf{h}_J) = \mathbf{h}_{J,\ell_1}$ in MACE).

## A.1  Graph Construction

**Node Features:**  For each molecular orbital $p$, an independent molecular graph is constructed where each node represents an atom. The node features for atom $A$ in the graph corresponding to orbital $p$ are simply the MO embeddings $\mathbf{x}_{pA}^{(0)} = \{x_{pA,k\ell m}^{(0)}\}$.

**Node Attributes:**  Each node stores information about atomic species as attributes. The atomic species for atom $A$ is encoded as a one-hot vector, $\mathbf{z}_A$, where each component corresponds to a different atomic element. This atomic-type information is essential for the model to distinguish between elements and their chemical properties. In the context of our model, it is important to understand the basis set.

**Radial weighting function:**  The edge features are computed using a radial embedding block that creates rich distance-dependent representations. Let $\mathbf{r}_A$ be the position of atom $A$, and $d_{AB} = ||\mathbf{r}_B - \mathbf{r}_A||$ be the interatomic distance between atoms $A$ and $B$. The radial embedding uses Bessel basis functions to encode the interatomic distances:

$$\mathbf{e}_{AB} = \text{RadialEmbedding}(d_{AB}), \tag{27}$$

where the radial embedding is computed as:

$$\text{RadialEmbedding}(r) = f_{\text{cutoff}}(r) \cdot \begin{bmatrix} \phi_1(r) \\ \phi_2(r) \\ \vdots \\ \phi_{N_{\text{basis}}}(r) \end{bmatrix}, \tag{28}$$

with Bessel basis functions:

$$\phi_n(r) = \sqrt{\frac{2}{r_{\text{max}}}} \cdot \frac{\sin(n\pi r/r_{\text{max}})}{r}, \tag{29}$$

and polynomial cutoff function:

$$f_{\text{cutoff}}(r) = \begin{cases} 1 - \frac{(p+1)(p+2)}{2}\left(\frac{r}{r_{\text{max}}}\right)^p + p(p+2)\left(\frac{r}{r_{\text{max}}}\right)^{p+1} - \frac{p(p+1)}{2}\left(\frac{r}{r_{\text{max}}}\right)^{p+2} & \text{if } r < r_{\text{max}} \\ 0 & \text{if } r \geq r_{\text{max}} \end{cases} \tag{30}$$

The direction vector $\mathbf{v}_{AB}$ is used separately in the MACE layer for computing spherical harmonics and directional features that are essential for equivariant message passing.

## B  Extended background on Correlated Methods

### B.1  Hartree-Fock

In the Hartree-Fock method, we assume that the many-electron wavefunction $\Phi$ can be represented by a simple product of one-electron wavefunctions, that are usually referred to as orbitals. Each orbital then contains exactly one electron and may extend over the whole molecule. However, because electrons are fermions, a simple product is not quite sufficient: indeed, when exchanging two electrons, the wavefunction should change sign. For that reason, the orbitals are arranged in a so-called Slater determinant, that ensures the correct behavior of the sign when exchanging particles. Physically, the Hartree-Fock approximation corresponds to treating each electron in the mean field generated by all the other electrons and nuclei, thereby neglecting the instantaneous electron-electron interactions that results in correlations between their positions.

The Hamiltonian is an operator that describes the energetic contributions of all particles inside a molecule, including particle-particle interactions and their kinetic energy. It can be written as:

$$\hat{H} = \hat{h} + \frac{1}{2}\sum_{ij} \widehat{r_{ij}^{-1}} \tag{31}$$

where the core Hamiltonian $\hat{h}$ contains the kinetic energy of all electrons, and the attraction between the fixed nuclei and all electrons, while $\widehat{r_{ij}^{-1}}$ is the Coulomb repulsion between electrons written in atomic units. To obtain the total energy of the system, we take the integral of the many-electron wavefunction $\Phi$ with the Hamiltonian over all electrons and all space, that can be informally written (for a real wavefunction):

$$E = \int \Phi(\mathbf{x})\hat{H}\Phi(\mathbf{x})d\mathbf{x} \tag{32}$$

where $\mathbf{x}$ denotes the coordinates of all the electrons, and the dependence on the nuclear positions $\mathbf{R}_A$ is implied for simplicity. To further simplify the notation, quantum chemists and physicists denote such integrals as an expectation value:

$$E = \langle\Phi|\,\hat{H}\,|\Phi\rangle \tag{33}$$

We can then replace $\Phi$ by the expression assumed in the Hartree-Fock approximation to obtain the Hartree-Fock energy:

$$E_{\text{HF}}[\{\psi_i\}] = \sum_i h_{ii} + \tfrac{1}{2} \sum_{ij} \langle ij \,||\, ji \rangle, \tag{34}$$

where $h_{ii} = \langle \psi_i \,|\, \hat{h} \,|\, \psi_i \rangle$ is a one-electron integral with the core Hamiltonian $\hat{h}$, and

$$\langle ij \,||\, ji \rangle = \langle \psi_i \psi_j \,|\, \widehat{r_{12}^{-1}} \,|\, \psi_i \psi_j \rangle - \langle \psi_i \psi_j \,|\, \widehat{r_{12}^{-1}} \,|\, \psi_j \psi_i \rangle \tag{35}$$

is an antisymmetrized two-electron integral with the interelectronic repulsion Coulomb operator. We expand on the meaning of these integrals in the next section, and focus on the Hartree-Fock method here.

Each orbital $\psi_p$ depends on the coefficients $\mathbf{C}$ as described in Equation 2.1. In the Hartree-Fock method, we aim to obtain the best possible orbitals by minimizing the Hartree-Fock energy as a function of the coefficients $\mathbf{C}$. This minimization yields the canonical Hartree-Fock in Equation 1, that we repeat here for convenience:

$$\mathbf{F}(\mathbf{C})\mathbf{C} = \varepsilon \mathbf{C} \tag{36}$$

The Fock matrix $\mathbf{F}$ contains one- and two-electron integrals similar to the ones that appear in the energy and therefore depends on the shape of the orbitals and the coefficients $\mathbf{C}$. Because of this dependence, the Hartree-Fock equations have to be solved iteratively, successively diagonalizing $\mathbf{F}$ to obtain updated $\mathbf{C}$ until the above equation is obeyed, at which point the columns of $\mathbf{C}$ are eigenvectors of $\mathbf{F}(\mathbf{C})$ and the final Hartree-Fock energy can be calculated.

## B.2 Integrals in quantum chemistry

As introduced above, integrals in quantum chemistry can be divided in two categories, one-electron integrals that are generally denoted by:

$$h_{ij} = \left\langle \psi_i \,|\, \hat{h} \,|\, \psi_j \right\rangle \tag{37}$$

and two-electron integrals that are denoted by:

$$\langle ij \,|\, ab \rangle = \langle \psi_i \psi_j \,|\, \widehat{r_{12}^{-1}} \,|\, \psi_a \psi_b \rangle \tag{38}$$

where, in general, indices $i$, $j$, $a$, $b$ can be different. Both of these integrals stem from integrals of many-electron wavefunctions over the Hamiltonian in the form of

$$\int \Phi(\mathbf{x}) \hat{H} \Phi'(\mathbf{x}) d\mathbf{x} = \langle \Phi | \, \hat{H} \, | \Phi' \rangle \tag{39}$$

as introduced in the previous section, but where now $\Phi$ and $\Phi'$ are not necessarily the same.

One-electron integrals only integrate over the coordinates of a single electron, and involve one orbital on each side of the operator $\hat{h}$. They can be written more explicitly as:

$$h_{ij} = \int \psi_i(\mathbf{r}|\{\mathbf{R}_A\}) \hat{h} \psi_j(\mathbf{r}|\{\mathbf{R}_A\}) d\mathbf{r} \tag{40}$$

In our case, each orbital is a combination of radial basis functions and spherical harmonics (see Equation 2.1) and the integral can be evaluated by standard quantum chemistry methods.

Two-electron integrals integrate over the coordinates of two electrons, and involve two orbitals on each side of the operator $\widehat{r_{12}^{-1}}$. Often, the integral $\langle ij \,|\, ab \rangle$ appears paired with the integral $- \langle ij \,|\, ba \rangle$, so for convenience the following antisymmetrized integral is introduced:

$$\langle ij \,||\, ab \rangle = \langle ij \,|\, ab \rangle - \langle ij \,|\, ba \rangle \tag{41}$$

Once again, a two-electron integral can be written more explicitly as:

$$\langle ij \,|\, ab \rangle = \int \psi_i(\mathbf{r}_1|\{\mathbf{R}_A\}) \psi_j(\mathbf{r}_2|\{\mathbf{R}_A\}) \frac{1}{\mathbf{r}_1 - \mathbf{r}_2} \psi_a(\mathbf{r}_1|\{\mathbf{R}_A\}) \psi_b(\mathbf{r}_2|\{\mathbf{R}_A\}) d\mathbf{r}_1 d\mathbf{r}_2 \tag{42}$$

where we expanded the operator $\widehat{r_{12}^{-1}}$ explicitly. Once again, all orbitals are composed of radial basis functions and spherical harmonics (see Equation 2.1) and the integrals are evaluated using standard quantum chemistry methods.

## B.3 Coupled Cluster

The coupled cluster method (Purvis III & Bartlett, 1982; Szabo & Ostlund, 2012) expresses the exact electronic wavefunction as an exponential excitation of a reference Hartree-Fock wavefunction:

$$|\Psi_{\text{CC}}\rangle = e^{\hat{T}} |\Phi_{\text{HF}}\rangle , \tag{43}$$

where the cluster operator $\hat{T} = \hat{T}_1 + \hat{T}_2 + \hat{T}_3 + \cdots$ generates single, double, triple, and higher excitations out of the reference state. Unlike Configuration Interaction (CI), which uses a linear expansion, this exponential form ensures size extensivity, the total energy scales correctly with the number of noninteracting subsystems, and includes higher-order excitation effects implicitly through products of lower-order terms (e.g., $\frac{1}{2}\hat{T}_1^2$). The coupled cluster equations are obtained by projecting the similarity-transformed Schrödinger equation,

$$\bar{H} |\Phi_{\text{HF}}\rangle = E_{\text{CC}} |\Phi_{\text{HF}}\rangle , \quad \text{with} \quad \bar{H} = e^{-\hat{T}} \hat{H} e^{\hat{T}}, \tag{44}$$

onto the reference and excited determinants. Truncating $\hat{T}$ to a given excitation rank defines a hierarchy of systematically improvable methods: Coupled Cluster with Singles and Doubles (CCSD) includes single and double excitations, CCSDT adds triples, CCSDTQ adds quadruples, and so on.

Coupled-cluster with single and double excitations (CCSD) is one of the most widely used correlated electronic-structure methods. In CCSD, the correlated wavefunction is written as

$$|\text{CCSD}\rangle = e^{\hat{T}_1 + \hat{T}_2} |\Phi_{\text{HF}}\rangle , \tag{45}$$

where

$$\hat{T}_1 = \sum_{ia} t_i^a \, \hat{a}_a^\dagger \hat{a}_i, \tag{46}$$

$$\hat{T}_2 = \tfrac{1}{4} \sum_{ijab} t_{ij}^{ab} \, \hat{a}_a^\dagger \hat{a}_b^\dagger \hat{a}_j \hat{a}_i, \tag{47}$$

and $|\Phi_{\text{HF}}\rangle$ is the Hartree-Fock reference determinant and $\hat{T}_1$ and $\hat{T}_2$ are the single- and double-excitation cluster operators. The unknown parameters of the method are the excitation amplitudes $\{t_i^a, t_{ij}^{ab}\}$, which specify the coefficients of these operators.

The amplitudes are determined by requiring that the projected Schrödinger equation is satisfied within the space of singly and doubly excited determinants. This leads to the nonlinear system of CCSD equations:

$$E_{\text{CCSD}} = \langle \Phi_{\text{HF}} | \hat{H} | \text{CCSD} \rangle , \tag{48}$$

$$0 = \langle \Phi_i^a | e^{-(\hat{T}_1 + \hat{T}_2)} \hat{H} | \text{CCSD} \rangle , \tag{49}$$

$$0 = \langle \Phi_{ij}^{ab} | e^{-(\hat{T}_1 + \hat{T}_2)} \hat{H} | \text{CCSD} \rangle , \tag{50}$$

where $|\Phi_i^a\rangle = \hat{a}_a^\dagger \hat{a}_i |\Phi_{\text{HF}}\rangle$ denotes a singly excited determinant and $|\Phi_{ij}^{ab}\rangle = \hat{a}_a^\dagger \hat{a}_b^\dagger \hat{a}_i \hat{a}_j |\Phi_{\text{HF}}\rangle$ denotes a doubly excited determinant.

Once a set of amplitudes is available, the CCSD energy can be evaluated directly as

$$E_{\text{CCSD}} = E_{\text{HF}} + \sum_{aibj} \left( t_{ij}^{ab} + t_i^a t_j^b \right) \left( 2 \langle ij|ab \rangle - \langle ij|ba \rangle \right). \tag{51}$$

This expression highlights a central feature of CCSD: the amplitudes serve as the effective parameters of a highly structured, physics-informed model that maps molecular orbital integrals to correlated energies.

### B.3.1    T1-TRANSFORMED INTEGRALS

Evaluation of the projected equations in Equation (48) requires a large number of tensor contractions. To simplify the notation and reduce computational cost, it is common to introduce the so-called *T1-transformed* one- and two-electron integrals,

$$\tilde{h}_{pq} = \sum_{rs}(\delta_{pr} - t_r^p)(\delta_{qs} + t_q^s)h_{rs}, \tag{52}$$

$$\widetilde{\langle pr|qs \rangle} = \sum_{tu}\sum_{mn}(\delta_{pt} - t_t^p)(\delta_{qu} + t_q^u)(\delta_{rm} - t_m^r)(\delta_{sn} + t_s^n)\langle tm|un \rangle. \tag{53}$$

These transformed integrals incorporate the effect of the single-excitation amplitudes and allow the CCSD equations to be expressed in a compact form.

Using this notation, the residual equations for the single and double amplitudes can be written as

$$0 = \sum_{ckd}(2t_{ki}^{cd} - t_{ij}^{cd})\widetilde{\langle ak|dc \rangle} - \sum_{ckl}(2t_{kl}^{ac} - t_{lk}^{ac})\widetilde{\langle kl|ic \rangle}$$

$$+ \sum_{ck}(2t_{ik}^{ac} - t_{ki}^{ac})\left[\tilde{h}_{kc} + \sum_l(2\widetilde{\langle kl|cl \rangle} - \widetilde{\langle kl|lc \rangle})\right]$$

$$+ \tilde{h}_{ai} + \sum_j(2\widetilde{\langle aj|ij \rangle} - \widetilde{\langle aj|ji \rangle}), \tag{54}$$

and

$$0 = \widetilde{\langle ab|ij \rangle} + \sum_{cd}t_{ij}^{cd}\widetilde{\langle ab|cd \rangle} + \sum_{kl}t_{kl}^{ab}\left(\widetilde{\langle kl|ij \rangle} + \sum_{cd}t_{ij}^{cd}\widetilde{\langle kl|cd \rangle}\right)$$

$$+ P_{ij}^{ab}\left(-\sum_{ck}\left[\frac{1}{2}t_{kj}^{bc}\left(\widetilde{\langle ka|ic \rangle} - \frac{1}{2}\sum_{dl}t_{li}^{ad}\widetilde{\langle kl|dc \rangle}\right) + t_{ki}^{bc}\left(\widetilde{\langle ka|jc \rangle} - \frac{1}{2}\sum_{dl}t_{lj}^{ad}\widetilde{\langle kl|dc \rangle}\right)\right]\right.$$

$$+ \frac{1}{2}\sum_{ck}\left(2t_{jk}^{bc} - t_{kj}^{bc}\right)\left[2\widetilde{\langle ak|ic \rangle} - \widetilde{\langle ak|ci \rangle} + \frac{1}{2}\sum_{dl}\left(2t_{il}^{ad} - t_{li}^{ad}\right)\left(2\widetilde{\langle lk|dc \rangle} - \widetilde{\langle lk|cd \rangle}\right)\right]$$

$$+ \sum_c t_{ij}^{ac}\left\{\left[\tilde{h}_{bc} + \sum_k\left(2\widetilde{\langle bk|ck \rangle} - \widetilde{\langle bk|kc \rangle}\right)\right] - \sum_{dkl}\left(2t_{kl}^{bd}\right)\widetilde{\langle lk|dc \rangle}\right\} -$$

$$\sum_k t_{ik}^{ab}\left[\tilde{h}_{kj} + \sum_l\left(2\widetilde{\langle kl|jl \rangle} - \widetilde{\langle kl|lj \rangle}\right) + \sum_{cdl}\left(2t_{lj}^{cd} - t_{jl}^{cd}\right)\widetilde{\langle kl|dc \rangle}\right]\right), \tag{55}$$

where $P_{ij}^{ab}$ is the permutation operator that enforces the required antisymmetry, $P_{ij}^{ab}A_{ij}^{ab} = A_{ij}^{ab} + A_{ji}^{ba}$. These expressions define the nonlinear mapping from amplitudes to residuals that must be driven to zero.

### B.3.2    SOLVING THE CCSD EQUATIONS AS A ROOT-FINDING PROBLEM

From a numerical perspective, CCSD can be interpreted as solving a high-dimensional nonlinear system

$$\boldsymbol{\Omega}(\mathbf{t}) = \mathbf{0}, \tag{56}$$

where $\mathbf{t}$ is the vector containing all $t_i^a$ and $t_{ij}^{ab}$ amplitudes and $\boldsymbol{\Omega}(\mathbf{t})$ denotes the collection of residual expressions given above.

Standard quantum-chemistry implementations solve this system iteratively using a quasi-Newton procedure. At iteration $n$, the amplitudes are updated according to

$$\mathbf{t}^{(n+1)} = \mathbf{t}^{(n)} - \boldsymbol{\varepsilon}^{-1}\boldsymbol{\Omega}(\mathbf{t}^{(n)}), \tag{57}$$

where $\boldsymbol{\varepsilon}$ is a diagonal approximation to the Jacobian matrix of the residual function. In practice, $\boldsymbol{\varepsilon}$ is constructed from orbital-energy differences and serves as a computationally inexpensive preconditioner.

B.3.3   QUASI-NEWTON SOLUTION ALGORITHM

The CCSD solver can be viewed as a specialized fixed-point iteration with a physics-motivated preconditioner. The procedure can be summarized algorithmically as follows.

---

**Algorithm 1** Quasi-Newton solution of the CCSD equations

---

**Require:** Hartree-Fock orbitals and integrals
**Ensure:** Converged amplitudes $\mathbf{t}^*$
 1: Initialize amplitudes $\mathbf{t}^{(0)}$ (typically zeros or MP2 estimates)
 2: Construct diagonal preconditioner $\varepsilon^{-1}$ from orbital energies
 3: **for** $n = 0, 1, 2, \ldots$ until convergence **do**
 4:     Evaluate residual vector
$$\mathbf{\Omega}^{(n)} = \mathbf{\Omega}(\mathbf{t}^{(n)})$$
 5:     **if** $\|\mathbf{\Omega}^{(n)}\| < \tau$ **then**
 6:         **break**                                                                                     *(converged)*
 7:     **end if**
 8:     Update amplitudes:
$$\mathbf{t}^{(n+1)} = \mathbf{t}^{(n)} - \varepsilon^{-1}\mathbf{\Omega}^{(n)}$$
 9: **end for**

---

## C   Data efficiency intuition

We are trying to give an intuitive motivation for why we believe an amplitude-predicting model can yield more data-efficient learning. For this, imagine two molecules, infinitely separated, such that they do not interact at all. An MLIP would get a single target energy label for the combined system (see Figure 5, top), while an amplitude-based model gets amplitudes that, if in localized orbital basis, spatially resolve the contributions of both subsystems individually, as schematically presented in Figure 5, bottom. If we overfit a model on the combined system and then ask for a prediction of the individual subsystems during inference, an MLIP that was only trained on the single target label has no way to properly attribute the energy to the individual subsystems, while an amplitude model could correctly predict the subsystem.

## D   Density matrix calculation

We can calculate any one particle observable given the 1-RDM $\gamma$. However, to calculate $\gamma$ at the CCSD level of theory requires the $\Lambda$ tensor, another $T_2$-like object. A common workaround for testing purposes is expectation value CCSD (XCCSD) (Korona & Jeziorski, 2006). In XCCSD(2), the 1-RDM is assembled without the $\Lambda$ tensor from the occupied-occupied block:

$$\gamma_{ij} = \delta_{ij} - \sum_a t_i^{a*} t_j^a - \frac{1}{2} \sum_{a,b,k} t_{ik}^{ab*} t_{jk}^{ab} \tag{58}$$

the virtual-virtual block:

$$\gamma_{ab} = \sum_i t_i^{a*} t_i^b + \frac{1}{2} \sum_{i,j,c} t_{ij}^{ac*} t_{ij}^{bc} \tag{59}$$

and the occupied-virtual blocks:

$$\gamma_{ia} = t_i^{a*} + \sum_{j,b} t_{ij}^{ab*} t_j^b \tag{60}$$

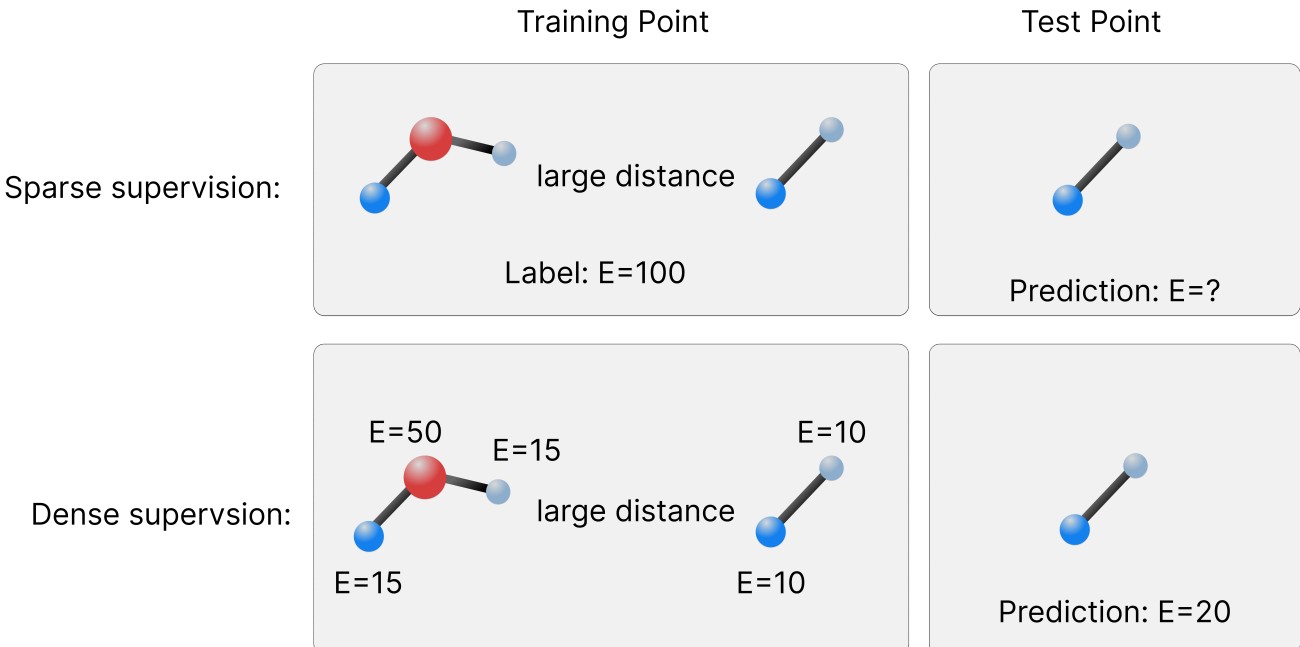

*Figure 5.* An MLIP that gets energy labels only for the combined system won't be able to predict subsystems correctly (top). In contrast, a model that gets dense supervision for all subsystems of the total system will be able to correctly predict individual components.

## E   Molecular orbital localization

Canonical molecular orbitals (MOs), as defined in Equation (1), are generally delocalized over the entire molecule. We can transform the occupied and virtual MOs by unitary rotations that mix orbitals only within their respective subspaces:

$$\tilde{\psi}_i = \sum_{j \in \text{occ}} (U_{\text{occ}})_{ij}\, \psi_j, \quad \mathbf{U}_{\text{occ}}^\dagger \mathbf{U}_{\text{occ}} = \mathbf{I} \tag{61}$$

$$\tilde{\psi}_a = \sum_{b \in \text{virt}} (U_{\text{virt}})_{ab}\, \psi_b, \quad \mathbf{U}_{\text{virt}}^\dagger \mathbf{U}_{\text{virt}} = \mathbf{I}. \tag{62}$$

We start by localizing the occupied orbitals with intrinsic bonding orbital (IBO) localization (Knizia, 2013). The IBO localization can be understood as a Pipek-Mezey (PM) localization (Pipek & Mezey, 1989), but with intrinsic atomic orbital (IAO) populations. Both PM and IBO localization minimize:

$$L = \sum_{A \in \text{atoms}} \sum_{i \in \text{occ}} (N_A(i))^m, \tag{63}$$

where $N_A(i)$ is the population of the $i$-th orbital on the $A$-th atom, and $m$ is the power typically chosen to be $2$ or $4$. The population $N_A(i)$ can be written as:

$$N_A(i) = \sum_{\mu\nu \in \text{basis centered on A}} C_{i\mu} C_{i\nu}, \tag{64}$$

where the labeling of the basis functions is the key difference between PM and IBO. IBO uses IAOs for assignment of basis function, as described by Knizia (2013).

For the virtual orbital localization, we have designed a localization scheme resembling that of Subotnik et al. (2005), later referred to as VV-HV (Wang et al., 2023). In this scheme, the valence virtuals are differentiated from the occupied ones by projecting onto a minimal basis, such as STO-3G or IAOs, which is then localized with IBO. This approach has strong resemblance and similarity to AVAS (Sayfutyarova et al., 2017). The leftover space of virtuals, i.e., hard virtuals, are constructed by projecting onto the original basis, shell by shell, and Schmidt orthogonalizing subsequent shells.

To recap, the localization scheme used is as follows: 1) IBO localize the occupied orbitals (with $m = 4$), 2) project the virtuals onto the IAO basis recovering valence virtuals, similar to AVAS, and localizing the whole set with IBO, and finally

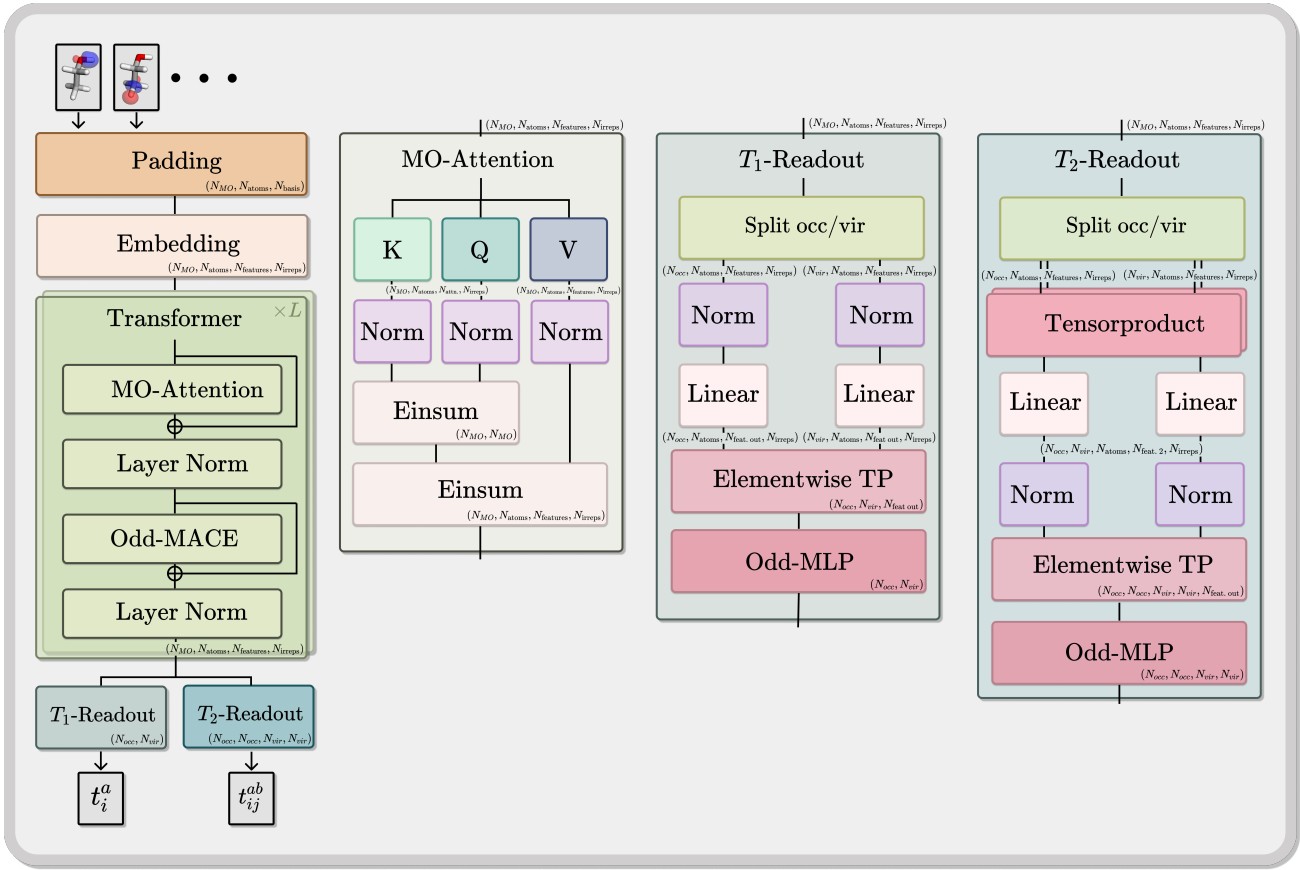

*Figure 6.* The individual components of the MōLe architecture.

3) project the leftover virtual space, shell by shell, symmetrically orthogonalizing of each shell, and removing those virtuals from subsequent operations.

Localized orbitals form an alternative representation of the occupied and virtual Hartree-Fock subspaces, leaving all observables, including the energy, invariant. This representation offers improved chemical interpretability, with features such as bonding and lone pairs more easily identified. Moreover, it yields sparse Fock, density, two-electron integrals, and, therefore, excitation-amplitude matrices. More importantly for us, we hypothesize that this representation facilitates better learning. To verify this intuition, we ablate localization and train a 64 feature model on canonical and localized orbitals on QM7. The results are in table 3. We can see that localization indeed cuts the error significantly.

*Table 3.* Ablation of the orbital localization. We train a smaller model with 64 irreps features in the transformer with canonical and localized orbitals. We see that localized orbitals reduce the error significantly.

| Model | QM7 |
| --- | --- |
| MōLe - canonical | 0.27 |
| MōLe - localized | **0.16** |

# F    Architecture details

In this sections we provide some more details about our MoLe architecture.

## F.1    Padding

Before encoding the MO coefficients, they have to be padded with zeros so that all atoms have the same number of MO coefficients. In irreps string notation, features of the form `ax0e + bx1e + cx2e + ...`, where the multiplicities

`a`,`b`,`c` depend on atom types and basis set, transform to `kx0e + kx1e + kx2e + ...`, with `k` being the maximum overall multiplicities of all atom types. The padded coefficients are then embedded into graphs by initializing the features of an equivariant graph neural network (see Figure 1).

## F.2   Layer Normalization

Let $x \in \mathbb{R}^{(L_{\max}+1)^2 \times C}$ be an irreps feature with maximum degree $L_{\max}$ and $C$ channels on an atom. We denote its components by $x_{m,k}^{(L)}$, where $L$ is the degree, $m \in \{-L, \ldots, L\}$ the order, and $k \in \{1, \ldots, C\}$ the channel index. The separable layer norm normalizes the scalar part ($L = 0$) and the rest ($L > 0$) separately:

**Scalar part** ($L = 0$).   We apply a standard layer normalization over channels to the scalar part $x_i^{(0)}$:

$$\mu^{(0)} = \frac{1}{C} \sum_{i=1}^{C} x_k^{(0)}, \quad \left(\sigma^{(0)}\right)^2 = \frac{1}{C} \sum_{k=1}^{C} \left(x_k^{(0)} - \mu^{(0)}\right)^2, \quad y_k^{(0)} = \gamma_k^{(0)} \frac{x_k^{(0)} - \mu^{(0)}}{\sigma^{(0)} + |\epsilon^{(0)}|}, \tag{65}$$

with learnable parameters $\gamma^{(0)}, \epsilon^{(0)} \in \mathbb{R}^C$.

**Higher-degree part** ($L > 0$).   All higher degrees $L \geq 1$ are normalized together as:

$$\left(\sigma^{(L)}\right)^2 = \frac{1}{C} \sum_{k=1}^{C} \frac{1}{2L+1} \sum_{m=-L}^{L} \left(x_{m,k}^{(L)}\right)^2, \quad \left(\sigma^{>0}\right)^2 = \frac{1}{L_{\max}} \sum_{L=1}^{L_{\max}} \left(\sigma^{(L)}\right)^2, \quad y_{m,k}^{(L)} = \gamma_k^{(L)} \frac{x_{m,k}^{(L)}}{\sigma^{>0} + |\epsilon^{>0}|}, \tag{66}$$

where each degree $L$ has learnable scale parameters $\gamma^{(L)}, \epsilon^{>0} \in \mathbb{R}^C$.

The output of the layer normalization block is then the concatenation over all degrees: $y = \{y_k^{(0)}\}_{k=1}^C \cup \{y_{m,k}^{(L)}\}_{L \geq 1, \, -L \leq m \leq L, \, 1 \leq k \leq C}$.

This construction is rotation-equivariant because it only rescales irrep blocks by scalar factors and does not mix orders $m$ within a given $L$. It also does not change the signs, thereby preserving sign equivariance. The $\epsilon$ ensures that, if all coefficients on an atom are 0 for a specific MO, they are also 0 after the normalization, ensuring size extensivity.

## F.3   Ablations

We perform ablations of the attention and Odd-MACE components in our architecture on a 64 and 128 feature model on the full QM7 training set. The results are in table 4. Both components are critical to achieve good performance.

*Table 4.* Validation loss for MōLe ablations. Removing the attention or Odd-MACE significantly worsens performance across the 64 and 128 feature model scales.

| Model | Validation loss |
|---|---|
| **64-width models** | |
| MōLe (64) | 33,828 |
| MōLe (64) – No Attention | 52,982 |
| MōLe (64) – No Odd-MACE | 126,699 |
| **128-width models** | |
| MōLe (128) | 17,131 |
| MōLe (128) – No Attention | 32,291 |
| MōLe (128) – No Odd-MACE | 85,072 |

# G   Visualization of results

## G.1   Amplitude visualization

We are visualizing the output of the $T_2$ amplitude in figure Figure 7. Since the $T_2$ amplitude is a four index tensor, we only plot one slice. We see that our model successfully predicts the highly non trivial structure of the amplitudes.

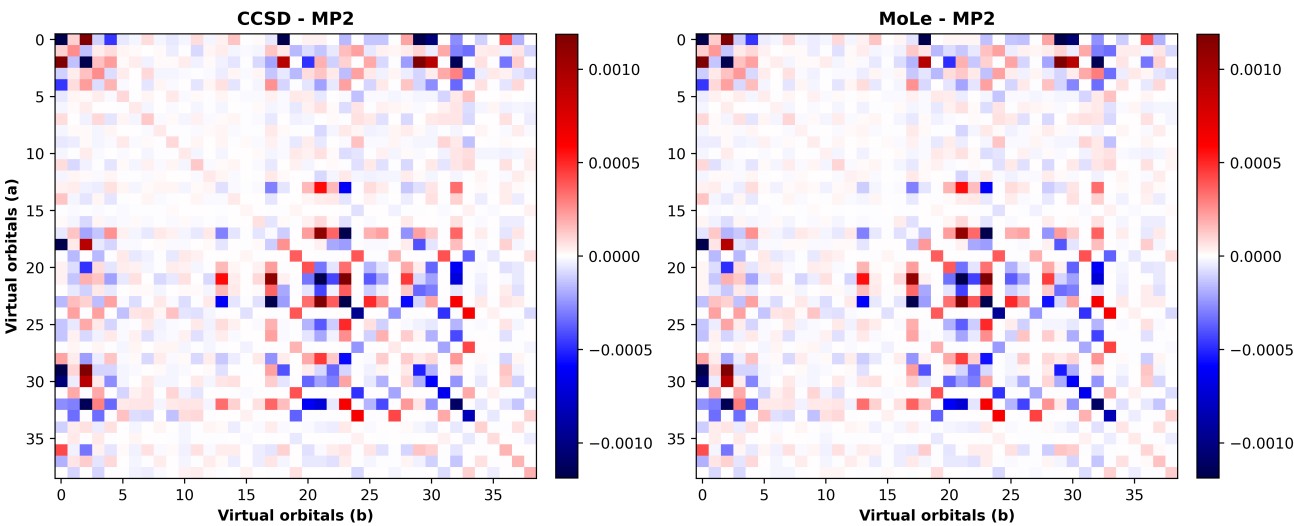

*Figure 7.* The prediction of MōLe and ground truth for the slice $(t_{\text{CCSD}} - t_{\text{MP2}})_{i=2,j=2}^{a,b}$ of Methanol. We see that MōLe correctly predicts the highly non-trivial correction between MP2 and CCSD.

### G.2 Time complexity scaling

In Figure 8, we time HF, MP2, CCSD (as implemented in GPU4PySCF (Li et al., 2024; Wu et al., 2024)), orbital localization, and MōLe for linear alkane chains from C13 to C21. HF, MP2, CCSD, and MōLe are run on a single H100 GPU, while localization is run on CPU. Over this finite size range, the fitted wall-time scaling is $\mathcal{O}(N^{5.9})$ for CCSD and $\mathcal{O}(N^{3.3})$ for MōLe. For the C17 alkane, the largest system before CCSD runs out of memory, MōLe is roughly 1000-fold faster than CCSD, while adding only a small overhead on top of the HF/MP2 calculation used to construct the baseline amplitudes.

## H CCSD solver iteration numbers

We provide the average number of CCSD solver iterations necessary to converge a CCSD calculation with default MP2 and MōLe initializations, as well as the number of systems that did not converge at all in Table 5. Clearly, the MōLe amplitudes serves as a high quality initial guess that can even make systems converge which would not have converged with MP2.

*Table 5.* The average CCSD cycles needed for convergence and number of unconverged systems with default MP2 vs. MōLe amplitudes. The convergence criterion is set to obtain an energy error of $\lesssim 0.1$ mHa compared to a very tightly converged calculation.

| Model | Amino Acids | | PubChem | |
|---|---|---|---|---|
| | Avg. cycles | Num. unconverged | Avg. cycles | Num. unconverged |
| MP2 (default guess) | 7.61 | 0 | 10.11 | 3 |
| MōLe | **3.83** | 0 | **6.3** | 1 |

## I Hyperparamters

We are training our MōLe models for 4 weeks on an H100 GPU using the Adam optimizer to minimize the mean squared error between the predicted and ground truth amplitudes:

$$\mathcal{L}(\{t_i^a\}, \{t_{i,\text{GT}}^a\}, \{t_{ij}^{ab}\}, \{t_{ij,\text{GT}}^{ab}\}) = \sum_{ia} \left( t_i^a - t_{i,\text{GT}}^a \right)^2 + \sum_{ijab} \left( t_{ij}^{ab} - t_{ij,\text{GT}}^{ab} \right)^2.$$

The hyperparameter details are given in table 6.

Mace is trained for 1000 epochs, which takes around 12h on a L40s GPU, using the recommended settings. One important change we did is to switch the default Adam to AdamW optimizer, which greatly improved results for MACE on our data.

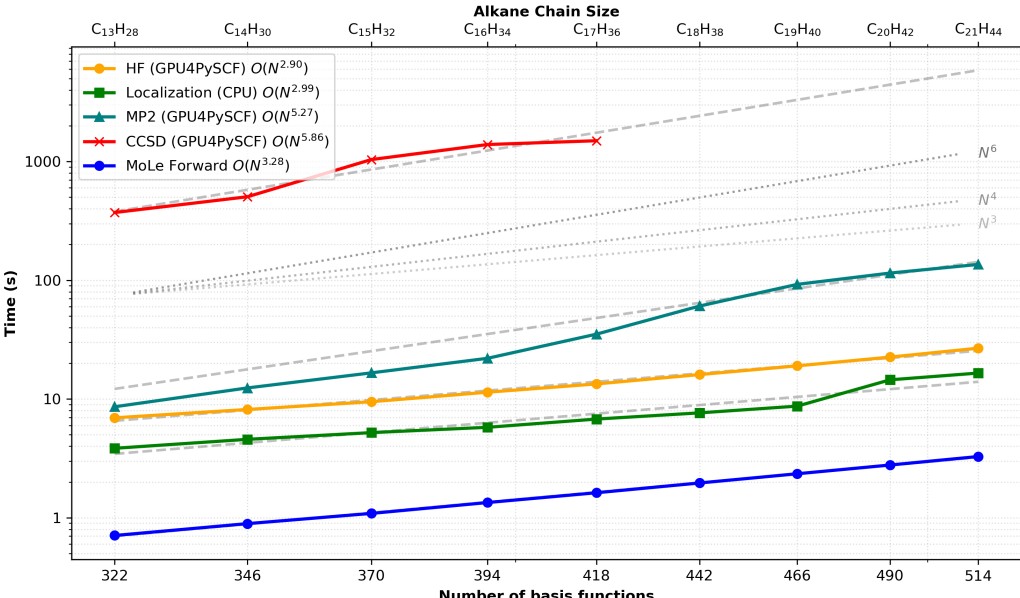

*Figure 8.* Timing of HF, MP2, and CCSD as implemented in GPU4PySCF ([Li et al., 2024](); [Wu et al., 2024]()) and MōLe for increasingly larger alkane systems. The localization code was run on CPU and timed similarly.

The eSEN model is based on the 6.3 million parameter small version, but with irreps higher than $\ell = 0, 1$ removed. Like MACE, we train eSEN for 1000 epochs, which takes around 10h on a L40s GPU.

| Category | Configuration |
|---|---|
| **Model / Geometry** | |
| Basis set | def2-SVP |
| Number of layers | 4 |
| **Transformer irrep dimensions** | |
| Hidden irreps | `128x0e + 128x1o + 128x2e` |
| Edge irreps | `1x0e + 1x1o + 1x2e` |
| **Radial / GNN Block** | |
| Radial type | Bessel |
| Number of Bessel functions | 10 |
| Polynomial cutoff order | 5 |
| Max radius | 4.0 |
| MACE correlation order | 3 |
| **Attention Block** | |
| Latent irreps | `32x0e + 32x1o + 32x2e` |
| Number of attention heads | 4 |
| **Readout Heads** | |
| $t_1$ readout irreps | `16x0e + 16x1o + 16x2e` |
| $t_1$ MLP neurons | 16 |
| Single$\rightarrow$Pair irreps | `16x0e + 16x0o + 16x1e + 16x1o + 16x2e + 16x2o` |
| Pair$\rightarrow$Quadruple irreps | `8x0e + 4x1e + 2x2e` |
| Pair$\rightarrow$Quadruple MLP neurons | 8 |
| Pair$\rightarrow$Quadruple layers | 1 |
| **Training** | |
| Batch size | 1 |
| Optimizer | Adam |
| Base learning rate | $10^{-2}$ |
| Loss function | MSE |
| Scheduler | StepLR |
| Scheduler step | 24 |
| Scheduler $\gamma$ | 0.5 |

*Table 6.* Model architecture and training hyperparameters used in our experiments.

| Category | Configuration |
|---|---|
| **Model Architecture** | |
| Number of interactions | 3 |
| Hidden irreps | 128x0e + 128x1o |
| Number of radial basis | 8 |
| Cutoff radius | 6.0 Å |
| **Training** | |
| Max epochs | 1000 |
| Batch size | 32 |
| Loss function | L1 (MAE) per atom |
| Gradient clipping | 10 |
| **Optimizer** | |
| Optimizer | AdamW |
| Learning rate | $10^{-3}$ |
| Weight decay | $10^{-3}$ |
| **LR Scheduler** | |
| Scheduler | Cosine Annealing + Warmup |
| LR min | $10^{-6}$ |

*Table 7.* Model architecture and training hyperparameters for MACE.

| Category | Configuration |
|---|---|
| **Model Architecture** | |
| Backbone | eSCN-MD |
| Number of layers | 4 |
| Hidden channels | 128 |
| Sphere channels | 128 |
| $\ell_{\max}$ | 1 |
| $m_{\max}$ | 1 |
| Edge channels | 128 |
| Distance function | Gaussian |
| Number of distance basis | 64 |
| Cutoff radius | 6.0 Å |
| Norm type | RMS norm (SH) |
| **Training** | |
| Max epochs | 1000 |
| Max atoms per batch | 256 |
| Loss function | L1 (MAE) per atom |
| Gradient clipping | 100 |
| **Optimizer** | |
| Optimizer | AdamW |
| Learning rate | $5 \times 10^{-4}$ |
| Weight decay | $10^{-3}$ |
| **LR Scheduler** | |
| Scheduler | Cosine |
| Warmup factor | 0.2 |
| Warmup epochs | 0.01 |
| LR min factor | 0.01 |

*Table 8.* Model architecture and training hyperparameters for eSEN.

