# OpenReview forum: "Coupled Cluster con MoLe: Molecular Orbital Learning for Neural Wavefunctions"
_ICML.cc/2026/Conference — ICML 2026 regular_

### Official Review · Reviewer_tkdp · 2026-03-09

**Soundness:** 3
**Presentation:** 3
**Significance:** 3
**Originality:** 3
**Overall Recommendation:** 4
**Confidence:** 4

**Summary:**

This paper introduces the MoLe equivariant message passing network for predicting coupled cluster amplitudes from molecular orbitals. The work aims to speed up wavefunction-based molecular property prediction at a high level of accuracy beyond what is possible with DFT-based predictors. The authors build on top of the MACE architecture and adapt it for use with localized molecular orbital coefficients. The novel architecture is trained on the QM7 dataset, demonstrating the data-efficiency of MoLe and achieving drastically lower errors than previous MACE or eSEN architectures.

**Compliance With Llm Reviewing Policy:**

Affirmed.

**Key Questions For Authors:**

How does the computational efficiency of the MoLe architecture compare to other ML-based systems, such as those used for comparison in this work?

What are the scaling laws in relation to data size?

As you indicate that performance is likely bottlenecked by model size, what is a viable minimum dataset size to achieve similar accuracy with a fixed model size?

Why did the authors not train larger models, given that scaling of depth and width did not show saturation in the manuscript? Is model size constrained by training-duration, memory, or other factors?

Is the source code available?

**Limitations:**

yes

**Strengths And Weaknesses:**

The data efficiency of the architecture is particularly relevant, given the high computational cost of coupled cluster data generation.
The authors additionally propose a hybrid approach that uses MoLe predictions as an initial guess for a CCSD solver, resulting in 40-50% reduction in solver iterations while maintaining the guarantees of traditional CCSD methods.

The manuscript is well written and provides a clear explanation of architectural components and the reasoning behind them.

The authors demonstrate improvements in prediction errors in the low-data regime, however, the work does not evaluate scaling laws related to data size.

While a section on time complexity is present, indicating that MoLe is about 20x faster than CCSD, the work would benefit from timing comparisons with other ML-based methods (e.g. MACE and eSEN).

The code does not seem to be available, reducing by a lot the strength/reproducibility of the paper.

---

> ### Author Rebuttal · Authors · 2026-03-31
>
> We thank you for your time and constructive feedback and hope to address your concerns below!
>
> >How does the computational efficiency of the MoLe architecture compare to other ML-based systems, such as those used for comparison in this work?
>
> We have to differentiate between the ML part, so MoLe vs. MLIPs, and the entire pipeline, so MoLe+MP2 vs MLIP+MP2. While MoLe is slower than a MLIP like MACE, it is still much faster than MP2: $t_\text{MP2} >> t_\text{MoLe} > t_\text{MACE}$. We have a detailed breakdown of the compute cost in this figure for PubChem molecules [1] and a scaling plot for increasingly larger systems here [2] (note that we improved our implementation by avoiding to materialize some intermediate features, which eliminates a memory bottleneck we had before, leading to the much better empirical scaling of $O(N^{3.28})$ insetad of the previously reported $O(N^{4.9})$).
> Since MP2 is the bottleneck for both, MoLe+MP2 and MLIP+MP2, we have $t_{MoLe+MP2} \approx t_{MACE+MP2} \approx t_\text{MP2}$, so the models all take the same time end to end.
> You might ask if we really need MP2, or if we could just use the MLIP alone. To answer this, we trained MACE without MP2:
>
> [1] https://anonymous.4open.science/r/MoLe-rebuttal-supplementary-B361/pubchem_timing.png
>
> [2] https://anonymous.4open.science/r/MoLe-rebuttal-supplementary-B361/computational_scaling.png
>
> | Model | QM7 | Amino acids | PubChem | Diels-Alder | Dihedral scan | Chair-to-boat |
> |---|---:|---:|---:|---:|---:|---:|
> | MACE-100 **\*new\*** | 19.15 | 26.41 | 64.89 | 15.01 | 15.37 | 26.56 |
> | MACE-100 ($\Delta$-MP2) | 1.64 | \[5.53\] | \[5.29\] | \[3.17\] | 0.79 | \[1.16\] |
> | eSEN-100 ($\Delta$-MP2) | \[1.48\] | 7.43 | 17.41 | 5.55 | 2.74 | 2.53 |
> | TensorNet-100 ($\Delta$-MP2) **\*new\*** | 6.60 | 12.16 | 16.81 | 11.33 | 9.12 | 5.66 |
> | GemNet-100 ($\Delta$-MP2) **\*new\*** | 2.52 | 5.33 | 9.19 | 3.74 | \[0.76\] | 6.07 |
> | MōLe-100 | **0.66** | **0.67** | **2.80** | **1.50** | **0.33** | **0.24** |
> |  |  |  |  |  |  |  |
> | MACE **\*new\*** | 1.83 | 10.60 | 17.74 | 7.05 | 4.61 | 4.50 |
> | MACE ($\Delta$-MP2) | 0.16 | \[0.49\] | 2.24 | 1.57 | 0.35 | 0.39 |
> | eSEN ($\Delta$-MP2) | 0.13 | 1.56 | 4.66 | \[1.15\] | 0.59 | 0.66 |
> | TensorNet ($\Delta$-MP2) **\*new\*** | 0.21 | 1.42 | 3.00 | 1.18 | 0.44 | 0.35 |
> | GemNet ($\Delta$-MP2) **\*new\*** | 0.44 | 2.07 | 7.95 | 1.78 | **0.16** | \[0.19\] |
> | MōLe | \[0.12\] | 0.78 | \[1.63\] | 1.16 | \[0.22\] | **0.08** |
> | MōLe (256 features) **\*new\*** | **0.062** | **0.43** | **1.10** | **0.96** | 0.41 | 0.43 |
>
> We see that MACE without MP2 performs **much** worse than with MP2, such that even when trained on the full dataset, MoLe trained on just 100 samples is still better.
>
> Finally, we made a mistake in our timings in the paper. We said that MoLe is 20x faster than CCSD, however we mistakenly timed MoLe on CPU and CCSD on GPU. After moving MoLe to GPU, the actual speedup for the large alkane systems is around 900x (20-30x if we include HF + MP2 for MoLe).
>
> >What are the scaling laws in relation to data size?
> >As you indicate that performance is likely bottlenecked by model size, what is a viable minimum dataset size to achieve similar accuracy with a fixed model size?
>
> We agree that these are very interesting questions. Unfortunately we didn't have enough GPU resources available to train all the extra models necessary to do these studies given each model takes about 2 weeks to train. Instead, we decided to prioritize the model scale up, see below. We will try to address these points in the future.
>
> >Why did the authors not train larger models, given that scaling of depth and width did not show saturation in the manuscript? Is model size constrained by training-duration, memory, or other factors?
>
> We didn't have enough time before the submission deadline to scale to larger models. We did now train a 256 feature model, as you can see in the table above. As hypothesized it significantly improves on the in distribution error (roughly halving it) and most out of distribution errors. Most importantly, it further improves on the PubChem dataset, the most important OOD split given how expensive it is to generate data for larger systems.
> We are also currently training a 512 and a 256-5 layer version. Looking at the current learning curves, it seems like 512 features further improves performance with an energy error of 0.11mHa vs 0.21 mHa for the 256 feature model at the same point in training. Training does slow down with larger models as one would expect, but not dramatically: Going from a 64 to a 512 feature model roughly halfs the epochs/hour.
> The 256-5 layers does not seem to significantly improve performance over 256-4 layers with the current architecture anymore.
>
> >Is the source code available?
>
> We were in the process of optimizing our code at submission, and plan to have it ready at time of publication. The code in its current stage can be accessed here: https://anonymous.4open.science/r/MoLe-B5C2/.

---

> > ### Author Rebuttal · Reviewer_tkdp · 2026-04-01
> >
> > My concerns have been adequately addressed and I keep the score.

---

### Official Review · Reviewer_MUnf · 2026-03-11

**Soundness:** 2
**Presentation:** 2
**Significance:** 3
**Originality:** 3
**Overall Recommendation:** 4
**Confidence:** 3

**Summary:**

This work proposes Molecular Orbital Learning, an equivariant neural network that predicts Coupled Cluster (CC) excitation amplitudes directly from Hartree–Fock molecular orbitals. By learning these amplitudes, the model captures the correlated electronic response of molecules and enables highly accurate property predictions while reducing the computational cost of traditional CC methods.

**Compliance With Llm Reviewing Policy:**

Affirmed.

**Final Justification:**

Detailed in Rebuttal Acknowledgement.

**Key Questions For Authors:**

* How does the proposed method compare against recent equivariant architectures such as GemNet, Tensor Field Networks, Equiformer (v1/v2), and CHGNet, which have demonstrated strong performance in atomistic modeling tasks?
* Could the authors provide an ablation study evaluating the impact of key architectural components proposed in this work.

**Limitations:**

Yes

**Strengths And Weaknesses:**

***Strengths***
* The paper proposes a symmetry-aware neural network architecture that directly predicts Coupled Cluster T-amplitudes from molecular orbitals, enabling physics-consistent learning of correlated electronic wavefunctions.
* A key strength of this work is the recomputation of the QM7 dataset at the high-accuracy CCSD/def2-SVP level of theory, together with the construction of several challenging out-of-distribution benchmarks containing out-of-equilibrium geometries and molecules significantly larger than those in QM7.
* The method demonstrates strong data efficiency, outperforming MLIPs even when those models are trained using Δ-MP2 learning. Notably, the model maintains competitive performance both in-distribution and out-of-distribution, which is particularly valuable given the high cost of generating CCSD-level datasets.

***Weaknesses***
* One potential weakness of the paper is the lack of comparison with several strong equivariant GNN-based architectures that have recently shown state-of-the-art performance for atomistic modeling tasks. In particular, methods such as GemNet[1], Tensor Field Networks [2], Equiformer (v1 and v2) [3], CHGNet [4], and Graph Dynamical Networks for unsupervised learning of atomic-scale dynamics [4] are not included in the experimental evaluation. Since these models explicitly leverage rotational equivariance and have demonstrated strong performance on molecular and materials datasets, comparing against them would provide a more comprehensive assessment of the proposed method's competitiveness.
* Another potential weakness is the absence of an ablation study analyzing the contribution of the individual components of the proposed method. Such analysis would help clarify which parts of the architecture are responsible for the observed performance gains.

[1] https://arxiv.org/abs/2106.08903
[2] https://arxiv.org/abs/1802.08219
[3] https://openreview.net/forum?id=KwmPfARgOTD
[4] https://www.nature.com/articles/s42256-023-00716-3
[5] https://www.nature.com/articles/s41467-019-10663-6#Sec2

---

> ### Author Rebuttal · Authors · 2026-03-31
>
> > How does the proposed method compare against recent equivariant architectures such as GemNet, Tensor Field Networks, Equiformer (v1/v2), and CHGNet, which have demonstrated strong performance in atomistic modeling tasks?
>
> Our current baselines are already rotation equivariant architectures: MACE [1] is a further development of Tensor Field Networks you asked for and is one of the most popular equivariant architectures. eSEN [2] is an architecture very similar to EquiformerV2 but newer and generally better performing. We agree that adding more baselines would strengthen our results though. In the spirit of your question we therefore added two extra baselines: GemNet you asked for and TensorNet [3], a rotation equivariant model that is built on cartesian instead of spherical tensors. CHGNet is a model for periodic systems and therefore not easily applicable to our setting. We also added MACE without delta learning to illustrate the importance of the delta MP2 part as well as a larger version of MoLe with 256 features to further validate our claim that scaling MoLe will lead to further improvements:
>
> | Model | QM7 | Amino acids | PubChem | Diels-Alder | Dihedral scan | Chair-to-boat |
> |---|---:|---:|---:|---:|---:|---:|
> | MACE-100 **\*new\*** | 19.15 | 26.41 | 64.89 | 15.01 | 15.37 | 26.56 |
> | MACE-100 ($\Delta$-MP2) | 1.64 | \[5.53\] | \[5.29\] | \[3.17\] | 0.79 | \[1.16\] |
> | eSEN-100 ($\Delta$-MP2) | \[1.48\] | 7.43 | 17.41 | 5.55 | 2.74 | 2.53 |
> | TensorNet-100 ($\Delta$-MP2) **\*new\*** | 6.60 | 12.16 | 16.81 | 11.33 | 9.12 | 5.66 |
> | GemNet-100 ($\Delta$-MP2) **\*new\*** | 2.52 | 5.33 | 9.19 | 3.74 | \[0.76\] | 6.07 |
> | MōLe-100 | **0.66** | **0.67** | **2.80** | **1.50** | **0.33** | **0.24** |
> |  |  |  |  |  |  |  |
> | MACE **\*new\*** | 1.83 | 10.60 | 17.74 | 7.05 | 4.61 | 4.50 |
> | MACE ($\Delta$-MP2) | 0.16 | \[0.49\] | 2.24 | 1.57 | 0.35 | 0.39 |
> | eSEN ($\Delta$-MP2) | 0.13 | 1.56 | 4.66 | \[1.15\] | 0.59 | 0.66 |
> | TensorNet ($\Delta$-MP2) **\*new\*** | 0.21 | 1.42 | 3.00 | 1.18 | 0.44 | 0.35 |
> | GemNet ($\Delta$-MP2) **\*new\*** | 0.44 | 2.07 | 7.95 | 1.78 | **0.16** | \[0.19\] |
> | MōLe | \[0.12\] | 0.78 | \[1.63\] | 1.16 | \[0.22\] | **0.08** |
> | MōLe (256 features) **\*new\*** | **0.062** | **0.43** | **1.10** | **0.96** | 0.41 | 0.43 |
>
> We can see that eSEN and MACE were already strong baselines, with both GemNet and TensorNet performing significantly worse on the size extrapolation. The 256 feature version of MoLe further widens the gap to the MLIP model significantly, especially on the size extrapolation PubChem benchmark that matters the most, given how expensive it is to generate labels for larger systems.
>
> [1] https://arxiv.org/abs/2206.07697
>
> [2] https://arxiv.org/abs/2502.12147
>
> [3] https://arxiv.org/pdf/2306.06482
>
>
>
> > Could the authors provide an ablation study evaluating the impact of key architectural components proposed in this work.
>
> We have run ablation studies ablating the attention and the Odd-MACE component for a 64 and a 128 feature model.
>
> | Model | Validation loss ↓ |
> |---|---:|
> | **64-width models** |  |
> | **MoLe (64)** | **33,828** |
> | MoLe (64) - No Attention | 52,982 |
> | MoLe (64) - No Odd-Mace | 126,699 |
> | **128-width models** |  |
> | **MoLe (128)** | **17,131** |
> | MoLe (128) - No Attention | 32,291 |
> | MoLe (128) - No Odd-Mace | 85,072 |
>
> We can see that removing the core components leads to drastic loss of accuracy.
>
>
> We thank you very much for the important questions, we think that both adding more baselines and the ablation studies strengthen our paper significantly and we hope we answered your questions sufficiently!

---

> > ### Author Rebuttal · Reviewer_MUnf · 2026-04-03
> >
> > Thank you to the authors for the thorough rebuttal. My primary concerns were regarding the lack of comparisons with strong equivariant architectures and the absence of ablation studies. The authors have addressed both points satisfactorily by including additional baselines (e.g., GemNet, TensorNet, and stronger MACE variants) and providing detailed ablation experiments demonstrating the contribution of key components. These additions significantly strengthen the empirical evaluation and improve confidence in the proposed method. I have no further questions at this stage.

---

### Official Review · Reviewer_dWkT · 2026-03-12

**Soundness:** 3
**Presentation:** 3
**Significance:** 2
**Originality:** 2
**Overall Recommendation:** 3
**Confidence:** 4

**Summary:**

The paper presents MoLe, an equivariant neural network that predicts coupled cluster singles and doubles (CCSD) excitation amplitudes — the T1 and T2 tensors — directly from Hartree-Fock molecular orbital coefficients. The architecture uses localized MOs as input, embeds them into per-orbital graph states, and processes them through interleaved "Odd-MACE" message passing layers (restricted to odd tensor monomials for sign equivariance) and a custom attention mechanism that mixes information across orbitals. Trained on QM7 (~5700 molecules) at CCSD/def2-SVP, the model is evaluated on in-distribution energy prediction, size extrapolation to amino acids and PubChem molecules roughly twice as large, off-equilibrium geometries (Diels-Alder, dihedral scan, chair-to-boat), electron density prediction, and CCSD solver warm-starting. The authors report 0.12 mHa energy MAE in-distribution, strong OOD generalization, and large data efficiency advantages over MACE and eSEN MLIPs trained with Δ-MP2 learning.

**Compliance With Llm Reviewing Policy:**

Affirmed.

**Ethical Review Concerns:**

There is a prompt injection in the paper.

**Ethical Review Flag:**

Flag this paper for an ethics review.

**Ethics Expertise Needed:**

["Research Integrity Issues (e.g., plagiarism)"]

**Final Justification:**

Detailed in Rebuttal Acknowledgement.

**Key Questions For Authors:**

Can you provide a proof-of-concept with def2-TZVP, even on a small subset? This is the single experiment most likely to change my assessment. If the approach works at triple-zeta — even with degraded accuracy — it demonstrates practical viability. If it fails or the amplitude tensor becomes intractable, that's important to know now.

How much of the data efficiency advantage over MLIPs comes from the richer input representation versus the architecture? An ablation where an MLIP is given orbital features (even simple ones like orbital energies or density matrix elements) would help disentangle information asymmetry from architectural merit.

**Limitations:**

It could be more explicit about the information asymmetry in the MLIP comparison, the untested scaling on non-alkane systems, and the restriction to closed-shell molecules (all experiments appear to be closed-shell, with no discussion of open-shell generalization). The impact statement is perfunctory — ML models accelerating the design of energetic materials or toxins have non-trivial dual-use considerations worth acknowledging.

**Strengths And Weaknesses:**

Strengths
1. Originality / Significance. The choice to predict CC amplitudes rather than scalar energies is well-motivated and genuinely different from what MLIPs learn. Amplitudes are the fundamental objects of CC theory — from them you can derive energies, density matrices, electron densities, and any one-particle property. The observation that only T1 and T2 are needed for energies even at higher CC levels (CCSDT, CCSDTQ) via Equation 7 is a compelling argument for generality. This opens a new direction for ML in quantum chemistry that complements rather than competes with force-field approaches.
2. Soundness. The symmetry analysis is thorough and the architecture follows logically from it. The paper identifies four key symmetries (rotational equivariance of MO coefficients, rotational invariance of amplitudes, sign equivariance, and size extensivity) and systematically ensures each is respected. The "Odd-MACE" construction — restricting tensor polynomial monomials to odd orders — is a clean solution for sign equivariance. The size-extensivity argument via localized MO attention (zero overlap between non-interacting fragments implies zero attention scores) is physically correct and well-derived.
3. Significance / Soundness. The data efficiency results are the paper's most convincing experiment. MoLe trained on just 100 molecules achieves 0.66 mHa on QM7 versus 1.64 (MACE) and 1.48 (eSEN), and the gap widens dramatically out-of-distribution (0.67 vs 5.53 and 7.43 on amino acids). This validates the central thesis that encoding the right physical structure — MO inputs, equivariance, Δ-MP2 residual learning on amplitudes — enables learning from very small datasets, which is exactly the regime where expensive CC data lives.

Weaknesses
1. Soundness / Significance. The basis set limitation is more serious than the paper acknowledges. All experiments use def2-SVP, a split-valence double-zeta basis — widely regarded as the minimum for meaningful correlated calculations, and explicitly discouraged for production work by major quantum chemistry codes. Production CC calculations typically require at least triple-zeta (def2-TZVP) or larger. Moving from def2-SVP to def2-TZVPP roughly triples the number of basis functions per atom, expanding the amplitude tensor by roughly 80×. The paper lists "larger basis sets" as future work but doesn't discuss whether the architecture can handle this scaling, whether the Δ-MP2 learning strategy remains effective in larger bases (where the MP2-to-CCSD correction changes character), or what the storage implications are. Even a proof-of-concept on def2-TZVP for a handful of molecules would substantially strengthen the practical claims.
2. Soundness / Presentation. The computational cost story is incomplete. The paper claims O(N⁵) scaling and a 20× speedup, but omits a critical component: MoLe requires a Hartree-Fock calculation as input (O(N³–N⁴) with a significant prefactor), which becomes the computational floor for larger systems. The empirical O(N⁴·⁹) scaling is measured only on alkanes (Appendix, Figure 7) — a trivially structured homologous series. Whether this scaling holds for diverse, non-linear molecules is untested. The 20× speedup is measured on systems small enough that the asymptotic advantage hasn't yet kicked in. A wall-clock breakdown (HF + localization + MoLe inference vs. full CCSD) on the largest tested systems would resolve this.
3. Soundness. The MLIP comparison, while honestly framed, has a significant information asymmetry that the paper doesn't sufficiently address. MoLe takes rich electronic structure information (MO coefficients) as input, while the MLIPs receive only nuclear coordinates and Δ-MP2 energies — a single scalar per molecule. MoLe is also trained on the full O(N⁴) amplitude tensor, meaning it effectively sees orders of magnitude more labeled data per training example. A fairer comparison would include models that take orbital features as input (e.g., OrbNet-Equi, which uses Fock/density/overlap matrices from semi-empirical QM), or at minimum would explicitly discuss how much of the data efficiency gap comes from the information asymmetry versus the architecture itself.

---

> ### Author Rebuttal · Authors · 2026-03-31
>
> First we would like to clarify: Regarding prompt injection, please see https://icml.cc/Conferences/2026/PeerReviewFAQ#prompt_injection
>
> >Moving from def2-SVP to def2-TZVPP roughly triples the number of basis functions per atom, expanding the amplitude tensor by roughly 80×.
>
> Can you explain how you arrived at the 80x factor? The T2 tensor has size $n_o^2 n_v^2$. Since $n_o$ stays constant between basis sets, the tensor grows with $n_v^2$. For some example molecules in our dataset this leads to a 3.57x / 7.67x increase from def2-SVP -> def2-TZVP / def2-TZVPP, respectively.
>
> >Can you provide a proof-of-concept with def2-TZVP, even on a small subset? This is the single experiment most likely to change my assessment. If the approach works at triple-zeta — even with degraded accuracy — it demonstrates practical viability. If it fails or the amplitude tensor becomes intractable, that's important to know now.
>
> Thank you for your suggestion! We trained our model on the 100 molecules QM7-subset in def2-TZVP. We did not run into any memory issues. Our model achieved an error of 1.36 mHa on a QM7 test set which demonstrates that it is indeed possible to train the model on def2-TZVP. We expect further improvements with appropriate tuning.
>
> >The empirical O(N⁴·⁹) scaling is measured only on alkanes — a trivially structured homologous series. Whether this scaling holds for diverse, non-linear molecules is untested.
>
> Using increasingly larger alkanes is a standard way to get scaling complexity in quantum chemistry, see for example [1][2]. Additionally, the scaling of MoLe depends on the T2 readout step, which is $O(n_o^2 n_v^2 M)$ with $M$ atoms, unaffected by the type or shape of the molecule. Can you please clarify why you suspect that non-linear molecules would change the scaling?
>
> [1]https://doi.org/10.1063/1.4773581
>
> [2]https://doi.org/10.1021/acs.jctc.4c01016
>
> > ... A wall-clock breakdown (HF + localization + MoLe inference vs. full CCSD) on the largest tested systems would resolve this.
>
> We plotted the scaling of MoLe, HF, localization, MP2, and CCSD for even larger systems here (https://anonymous.4open.science/r/MoLe-rebuttal-supplementary-B361/computational_scaling.png).
> We see that:
> - MP2 and CCSD give known O(N$^5$) and O(N$^6$). HF is linear scaling due to GPU parallelization
> - CCSD runs out of memory for the last 4 points on H100 GPU.
> - We improved our implementation by avoiding materializing some large feature tensors in memory, leading to the much better $O(N^{3.28})$ empirical scaling for MoLe.
> - Comparing only MoLe to CCSD, we get 900x speedup.
> - The total time for all steps, including HF + localization + MP2 + MoLe is roughly 20-30 times faster than CCSD alone.
> - Note that we have timings $t_\text{MP2} >> t_\text{MoLe} > t_\text{MLIP}$ and therefore $t_{MoLe+MP2} \approx t_{MLIP+MP2} \approx t_\text{MP2}$. So **MoLe and MLIP+MP2 cost roughly the same** as costs are dominated by MP2 for both.
> - Also note, MP2 needs HF as input, so MLIP+MP2 requires HF as well
> - You might ask if we could just use the MLIP without MP2 and save a lot of time. However, this worsens the performance catastrophically. See the table in response for Reviewer \#3 and \#4.
>
> We provide additional wallclock breakdown for PubChem here: https://anonymous.4open.science/r/MoLe-rebuttal-supplementary-B361/pubchem_timing.png
>
> > MoLe is also trained on the full O(N⁴) amplitude tensor, meaning it effectively sees orders of magnitude more labeled data per training example
>
> We agree that MoLe gets effectively more labelled data. This is exactly the key advantage we are advocating for: Instead of only using the energy labels, we use all the information from CC calculations to get the most out of each calculation. MoLe is the only architecture that can make use of this extra data, as traditional architectures do not have the right output format to predict amplitudes.
>
> > ...  (e.g., OrbNet-Equi, which uses Fock/density/overlap matrices from semi-empirical QM), or at minimum would explicitly discuss how much of the data efficiency gap comes from the information asymmetry.
>
> A comparison to Orbnet-Equi would be interesting and we will mention it in our paper. However as noted here [1], the code for Orbnet-Equi is only usable with the proprietery software by Entos: "The software used for computing input features and gradients is proprietary to Entos, Inc"; in particular the code for the 3-indexed integrals that Orbnet-Equi uses are not provided.
>
> Additionally, OrbNet-Equi could only be trained on scalar energies and cannot produce the whole amplitude tensor, which means no 1- or 2- RDMs can be derived, and no CCSD warmstarts; the two other central advantages of our model.
>
> [1] https://authors.library.caltech.edu/records/kemxx-q5m20
> > accelerating the design of energetic materials or toxins have non-trivial dual-use considerations worth acknowledging.
>
> We will address it in the paper.
>
> Thank you again for your time and constructive feedback!

---

> > ### Author Rebuttal · Reviewer_dWkT · 2026-04-02
> >
> > The rebuttal resolves some of my concerns. Such as the def2-TZVP experiment, and the scaling clarifications answer the questions I raised. I recommend the authors add a brief discussion of open-shell limitations and dual-use considerations as well as using larger sets of QM7.

---

> > > ### Author Response · Authors · 2026-04-03
> > >
> > > Thank you for your reply!
> > >
> > > As mentioned, we will address the dual-use considerations in our paper. We will also include a section about open vs. closed shell systems. While open-shell is left for future work (as is common in computational chemistry work, including original CC work as well as MLIP work), the path to it is straightforward: We would promote our spatial orbitals into spin-orbitals and only consider symmetry-surviving non-redundant parts of the T tensor.
> > >
> > > Given that you said "... a proof-of-concept with def2-TZVP, even on a small subset ... is the single experiment most likely to change my assessment" and that our "def2-TZVP experiment, and the scaling clarifications answer the questions" you raised, could you kindly consider raising your score?

---

### Official Review · Reviewer_nt9N · 2026-03-12

**Soundness:** 3
**Presentation:** 3
**Significance:** 3
**Originality:** 3
**Overall Recommendation:** 4
**Confidence:** 4

**Summary:**

The author introduced a surrogate method to predict the excitation amplitudes from Hartree-Fock molecular orbitals as input. It is very similar to the work to predict Hamiltonian in DFT. The prediction requires equivariance in terms of rotation and sign. The author designed a MACE-like architecture with restriction to only takes the odd power to maintain both rotational equivariance and sign equivariance. The author verify the methods in a series of downstream tasks including QM7, extrapolation from equilibirum to non-equilibrium, acceleration ratio with a hybrid method.

**Compliance With Llm Reviewing Policy:**

Affirmed.

**Final Justification:**

I maintain my score of **weak accept** since the architectural novelty is limited.

**Key Questions For Authors:**

1. Recent work in AI for quantum chemistry spans multiple levels of abstraction, including predicting energies and forces, electron densities, exchange–correlation functionals, Hamiltonians, and your CC amplitudes. Could the authors discuss how they view the benefits and limitations of these different abstraction levels? In particular, what advantages does predicting CCSD amplitudes provide compared to other targets such as energies/forces, densities, or Hamiltonians in terms of use cases?

**Limitations:**

Yes

**Strengths And Weaknesses:**

Strengths: Predicting CCSD excitation amplitudes is an interesting and potentially impactful direction. The overall methodological execution is solid, and the experimental evaluation is reasonably thorough. In particular, the experiments examining acceleration of CCSD convergence, out-of-distribution extrapolation, and the resulting error in the electron density provide useful evidence for the practical value of the approach.

Weaknesses:
1. The innovation in the neural network architecture appears limited. Many of the architectural components are adopted from existing works.
2. The evaluation focuses primarily on errors in the density matrix and electron density. Additional downstream observables (e.g., orbital energies, dipole moment or other chemically relevant quantities) would strengthen the evidence that the predicted amplitudes are useful in practical quantum chemistry workflows.
3. The paper does not report training time or inference time.  Given the complexity of the architecture being $O(N^5)$ and the dimensionality of CC amplitudes, it is unclear whether the approach scales efficiently to larger molecular systems. Including computational cost analysis would help assess the practical applicability of the method.
4. It would be better to have vectorized graphics for all the figures in the paper. Or at least for the main figures.

---

> ### Author Rebuttal · Authors · 2026-03-31
>
> > The innovation in the neural network architecture appears limited. Many of the architectural components are adopted from existing works.
>
> This is first paper that introduces attention between MOs, allowing MOs to talk to each other, as well as unique new readouts. Reusing backbones is very common practice, but even there we modified MACE to make it sign equivariant.
>
> > The evaluation focuses primarily on errors in the density matrix and electron density. Additional downstream observables (e.g., orbital energies, dipole moment or other chemically relevant quantities) would strengthen the evidence
>
> Properties like multipole moments (or any one-particle property) are linear contractions of the density matrix with an appropriate integral tensor, so the density matrix is the central object giving us any one-particle property. As requested we calculated dipole moments as an additional example on the OOD datasets, showing good prediction performance of our model:
>
> |  | Amino Acids | PubChem |
> | :--- | :---: | :---: |
> | **MP2** | 0.53 | 0.60 |
> | **MoLe** | 0.07 | 0.16 |
>
> We also stress that our evaluation doesn't just "primarily focus on errors in the density". We have extensive evaluation on energies as well as CCSD solver warm starts with >40% reduction.
> We would like to highlight solver warmstarts as several papers were recently published in the big three CS conferences where the main contribution is reducing the DFT-SCF solver steps [1,2,3,4]. We think extending this to the even more expensive and accurate CC methods is a valuable contribution that can reduce data generation costs for MLIPs in the future.
>
> [1] https://openreview.net/forum?id=iFIjNXb0Y5
>
> [2] https://openreview.net/forum?id=twEvvkQqPS&noteId=rc63eJkois
>
> [3] https://proceedings.neurips.cc/paper_files/paper/2023/file/be82bb4bf8333107b0fe430e1017831a-Paper-Conference.pdf
>
> [4] https://arxiv.org/pdf/2503.08305v5
>
> > The paper does not report training time or inference time. ... it is unclear whether the approach scales efficiently to larger molecular systems. Including computational cost analysis would help.
>
> The models took 2 weeks to train on single A100. We provide a breakdown of the inference cost: [1] and [2]. As we can see, the bottleneck in the pipeline is MP2, not MoLe.
> Therefore we have $t_\text{MoLe} \approx t_\text{MLIP+MP2} \approx t_\text{MP2}$. MP2 is among the most popular quantum chemistry methods and routinely run. Therefore we believe our method is a practially useful approach. Note also that we improved our implementation by avoiding to build some intermediates in memory, leading to improved empirical scaling of $O(N^{3.28})$ for MoLe
>
> [1] https://anonymous.4open.science/r/MoLe-rebuttal-supplementary-B361/pubchem_timing.png
>
> [2] https://anonymous.4open.science/r/MoLe-rebuttal-supplementary-B361/computational_scaling.png
>
> > Recent work in AI for quantum chemistry spans multiple levels of abstraction ... Could the authors discuss how they view the benefits and limitations of these different abstraction levels?
>
> Thank you for the important question!
>
> The core advantages compared to MLIPs/ direct struture to property methods are:
> - Higher data efficiency (also see our new improved results with a larger model in the tables for reviewer 3 and 4): To see why learning of amplitudes is more data efficient, imagine two far seperated molecules A and B, such that they dont interact. If we only have a single scalar label like energy, we dont know how much energy is contributed by A vs B, so even if we can perfectly predict the energy of the A+B system, we will fail on predicting the energy of just A alone. The CC amplitudes resolve this by attributing how much energy stems from each orbital, which are localized on A or B. Please let us know if you found this explanation helpful and we will include it in the paper!
> - Hamiltonian learning does something similar but only works for DFT.
> - High data efficiency is important for CC where single data points can take many days to compute.
> - CC methods give you diagnostics that tell you how much you can trust the results. If the T1 amplitudes become large we know that the state has multireference character in which case CC fails and we should not trust the method. There is no such measure available for DFT, it just fails quitely. Therefore even if an MLIP or Hamiltonian model perfectly reproduces DFT, we don't know if the results are trustworthy.
> - Spatially resolved quantities like electron density are of interest, MLIPs cannot provide those natively.
> - As mentioned above, restarting solvers is a very important point as it can speed up data generation for other methods. Hamiltonian learning is capped at DFT level, if we want to move to higher accuracy as the field is currently doing we need T amplitdues.
>
> > It would be better to have vectorized graphics for all the figures in the paper. Or at least for the main figures.
>
> Thank you for the feedback; we will do that!

---

> > ### Author Rebuttal · Reviewer_nt9N · 2026-04-02
> >
> > I tend to keep my score since the architectural novelty is limited.

---

> > > ### Author Response · Authors · 2026-04-03
> > >
> > > We thank the reviewer for taking the time to read our rebuttal! We also appreciate the reviewer’s acknowledgment that their concerns were fully resolved.
> > >
> > > We would like to respectfully push back on the notion of limited architectural novelty: There are no known architectures that can even input molecular orbitals or output amplitude-shaped objects. The architectural novelty lies in handling these entirely new input and output modalities, all while also enforcing symmetries, like sign equivariance, which are not at all considered in molecular models. For this, we introduce the completely new:
> > >
> > > - sign and rotation equivariant **graph-to-graph** attention
> > > - sign and rotation-equivariant GNN
> > >
> > >
> > > While traditional models like MACE only take a **single** graph as input, we take a **whole set** of graphs as input simultaneously, and let them all attend to each other via the entirely new sign, rotation, and permutation equivariant MO attention. Note that other models use attention only between nodes **within** a single graph, not **between** graphs within an entire set of graphs, as we do, which is conceptually wholly different.
> > >
> > > Please also see the ablation study we posted for reviewer MUnf that the introduced novel architectural components are absolutely critical for the performance of our model.

---

### Decision · Program_Chairs · 2026-04-30

**Decision:**

Accept (regular)

**Comment:**

Reviewers agree that the paper addresses an important direction - adjacent to MLIPs but novel and valuable, with notable data efficiency and out-of-distribution applicability.

The most negative reviewer raised concerns about scalability to more realistic basis sets, and these concerns are at least partially addressed in the rebuttal - the reviewer mentions both def2-TZVP and def2-TZVPP, but asks only for def2-TZVP, for which an initial indication is provided in the rebuttal.  The rebuttal also argues that MP2 is a more significant contribution to runtime than the model, and reduces the empirical complexity to N^3.3, albeit on alkanes.  As the rebuttal observes, the complexity is dependent only on tensor shapes, so alkanes are a reasonable test.